# GEO4PALM v1.1: an open-source geospatial data processing toolkit for the PALM model system

Dongqi Lin[1], Jiawei Zhang[2], Basit Khan[3,a], Marwan Katurji[1], and Laura E. Revell[4]

[1]Te Kura Aronukurangi / School of Earth and Environment, University of Canterbury, Ōtautahi / Christchurch, New Zealand
[2]Scion, New Zealand Forest Research Institute Limited, Ōtautahi / Christchurch, New Zealand
[3]Geoinformatics for Climate Resilient Urban Systems (GRUSS), Institute of Photogrammetry and Remote Sensing (IPF), Karlsruhe Institute of Technology (KIT), 76131 Karlsruhe, Germany
[4]Te Kura Matū / School of Physical and Chemical Sciences, University of Canterbury, Ōtautahi / Christchurch, New Zealand
[a]Now at Arabian Center for Climate and Environmental Sciences (ACCESS), New York University Abu Dhabi, Abu Dhabi, United Arab Emirates

**Correspondence:** Dongqi Lin (dongqi.lin@canterbury.ac.nz)

**Abstract.** A geospatial data processing tool, GEO4PALM, has been developed to generate geospatial static input for the Parallelised Large Eddy Simulation (PALM) model system. PALM is a community-driven large eddy simulation model for atmospheric and environmental research. Throughout PALM's 20-year development, research interests have been increasing in its application to realistic conditions, especially for urban areas. For such applications, geospatial static input is essential. Although abundant geospatial data are accessible worldwide, geospatial data availability and quality are highly variable and inconsistent. Currently, the geospatial static input generation tools in the PALM community heavily rely on users for data acquisition and pre-processing. New PALM users face large obstacles, including significant time commitments, to gain the knowledge needed to be able to pre-process geospatial data for PALM. Expertise beyond atmospheric and environmental research is frequently needed to understand the data sets required by PALM. Here, we present GEO4PALM, which is a free and open-source tool. GEO4PALM helps users generate PALM static input files with a simple, homogenised, and standardised process. GEO4PALM is compatible with geospatial data obtained from any source, provided that the data sets comply with standard geo-information formats. Users can either provide existing geospatial data sets or use the embedded data interfaces to download geo-information data from free online sources for any global geographic area of interest. All online data sets incorporated in GEO4PALM are globally available, with several data sets having the finest resolution of 1 m. In addition, GEO4PALM provides a graphical user interface (GUI) for PALM domain configuration and visualisation. Two application examples demonstrate successful PALM simulations driven by geospatial input generated by GEO4PALM using different geospatial data sources for Berlin, Germany and Ōtautahi / Christchurch, New Zealand. GEO4PALM provides an easy and efficient way for PALM users to configure and conduct PALM simulations for applications and investigations such as urban heat island effects, air pollution dispersion, renewable energy resourcing, and weather-related hazard forecasting. The wide applicability of GEO4PALM makes PALM more accessible to a wider user base in the scientific community.

# 1 Introduction

The Parallelised Large Eddy Simulation (PALM) model is a Large Eddy Simulation (LES) model that has been used for Atmospheric Boundary Layer (ABL) research for over 20 years (Maronga et al., 2015, 2020; Raasch and Schröter, 2001). PALM is a free and open-source model with high scalability to simulate atmospheric flows from the mesoscale (10 to 200 km) to the microscale (1 cm to 1 km). To resolve land surface physics, PALM provides several features including the Radiative Transfer Model (RTM; Krč et al., 2021), Land Surface Model (LSM; Gehrke et al., 2021), the Urban Surface Model (USM; Resler et al., 2017), and the Plant Canopy Model (PCM; Maronga et al., 2020). Over the last few years, PALM has been extensively developed for various microscale and mesoscale applications, especially for wind energy and urban applications. With the implementation of new modules, PALM has been used to study wind turbine wake in a German wind farm (Vollmer et al., 2017) and urban environments, such as ventilation in the city of Hong Kong (Gronemeier et al., 2017), urban air quality and pollutant dispersion in Cambridge, United Kingdom (Kurppa et al., 2019), Helsinki, Finland (Kurppa et al., 2020), and Bergen, Norway (Wolf et al., 2020, 2021). In addition, several studies (e.g. Belda et al., 2021; Resler et al., 2021; Salim et al., 2022) have carried out sensitivity analysis and simulations to validate PALM in urban environments. Geospatial data sets that describe the land-surface characteristics and provide ground surface boundary conditions to the atmospheric model are critical for realistic simulations, especially for microscale and/or urban climate studies.

The exchanges of energy and moisture between the surface and the atmosphere are impacted by physical characteristics of the land surface, such as urban canopy, plant canopy, topography and land use, across a spectrum of spatial and temporal scales throughout the atmospheric boundary layer (ABL; e.g. Bou-Zeid et al., 2004; Maronga et al., 2014; Rihani et al., 2015; Srivastava et al., 2020). To include, resolve, and realise the near-surface physical characteristics, PALM's current initialisation setup allows users to provide a NetCDF static driver strictly formatted in PALM Input Data Standard (PIDS) as an input (hereafter referred to as the static driver). In PALM, geospatial data should be processed and stored in a static driver to carry out simulations. Heldens et al. (2020) described the data requirements of the static driver for PALM and provided the PALM Create Static Driver tool (hereafter PALM CSD) to generate a static driver for PALM using geospatial data. However, the data processing routine provided by Heldens et al. (2020) is heavily dependent on the geospatial data set prepared by the German Space Agency (DLR), for example, for three cities in Germany (Stuttgart, Berlin, and Hamburg) described in their study. The PALM CSD tool can only process data in NetCDF format with its particular data standard, which requires users to dedicate significant time to pre-processing the geospatial data.

In addition to PALM CSD, palmpy (https://github.com/stefanfluck/palmpy; Fluck, 2020) is another tool developed to generate static driver input for PALM applications at the Center for Aviation (ZAV), Zurich University of Applied Sciences, Switzerland (Liu et al., 2022). The palmpy tool is more generally applicable compared to PALM CSD, but, to the best of our knowledge, it has mainly been applied to regions in Switzerland. Another tool for PALM static drivers is the PALM-4U GUI developed at Fraunhofer Institute for Building Physics (https://gitlab.cc-asp.fraunhofer.de/palm_gui/palm4u_gui; last access: 7 November 2023). This tool, however, was only recently made public, and its user manual is still under construction at the time of writing. Users are responsible for a significant amount of data pre-processing before using these tools. Therefore, in

many other regions, for instance, New Zealand, where only a small number of geospatial data sets have been prepared by local authorities, big hurdles still exist to apply PALM with realistic land surface characteristics. Numerous geospatial data sets can be used to generate the PALM static driver, while the spatial coverage, resolution, data quality, and data format could vary. PALM simulations for urban applications may require high-resolution geospatial input, but PALM can also be used for applications of a coarse resolution over a large area. Users are required to search and identify the appropriate geospatial data sets for their PALM applications. In addition, the final conversion to PALM-readable formats requires extra processing. These issues go beyond the understanding and knowledge of physical processes and may have prevented further applications of PALM in the community. Furthermore, the lack of a highly applicable static driver preparation tool likely hinders the reproducibility of scientific results across different regions and research groups.

Looking at the history of Numerical Weather Prediction (NWP) models, the Weather Research and Forecasting (WRF) model is arguably one of the most popular numerical atmospheric models in the world, and its broader development has been community-driven (Powers et al., 2017). The community effort has empowered WRF users towards more advanced research and operational applications. In the WRF community, tools and packages have been developed with community contributions (e.g. as cited in Meyer and Riechert, 2019; Powers et al., 2017, and references therein). In contrast, the supporting tools for the PALM community are still limited. As discussed in Maronga et al. (2020), more development of PALM is still needed to broaden its applicability and accessibility and to strengthen its position within the boundary layer and urban climate scientific community. Through developing accessible and user-friendly tools with continuous efforts from the community, PALM has the opportunity to become as popular and widely used as WRF.

We have developed a widely applicable geo-information toolkit containing a set of routines written in the Python programming language designed to process geospatial data for PALM simulations. This tool is hereafter referred to as GEO4PALM. GEO4PALM can interface with several free online application programming interfaces (APIs), allowing users to obtain domain-specific information from globally available databases. For users who have obtained their geospatial data from other sources, GEO4PALM can process any geospatial data in GeoTIFF format regardless of the data source. With GEO4PALM, users can include land surface characteristics such as topography, land use, and building and plant canopy information in their PALM simulations. Here, we describe and document the GEO4PALM toolkit. The PALM static driver features covered by GEO4PALM are described in Section 2. Along with the framework and workflow of GEO4PALM, Section 3 presents detailed descriptions and requirements of the input data and the online data interfaces for GEO4PALM. Two application examples of GEO4PALM are given in Section 4. Conclusions, discussions on the limitations, and an outlook of GEO4PALM are presented in Section 5.

## 2  PALM features in GEO4PALM

The hierarchy and data format of the variables in the static driver of PALM are described in the PIDS (https://palm.muk.uni-hannover.de/trac/wiki/doc/app/iofiles/pids, last access: 20 June 2023) and by Heldens et al. (2020). Depending on the application, PALM simulations can include surface features such as buildings, pavements, and plants. In GEO4PALM, two settings

are available for users to choose from. One is the default or the minimum setting, and the other allows users to add additional features for the urban surface and plant canopies. For the minimum setting, the static driver is incorporated with all the required

variables to conduct a PALM simulation, while all the additional features are optional. Although GEO4PALM does not cover all the available features presented in PIDS, it includes most of the available features, including basic urban features such as buildings, pavements, and streets, which are sufficient to represent most urban and plant canopies. The variables covered by GEO4PALM are presented in Table 1.

**Table 1.** List of variables GEO4PALM can include in the PALM static driver. For more detailed descriptions of the variables, refer to Heldens et al. (2020) and the PIDS (https://palm.muk.uni-hannover.de/trac/wiki/doc/app/iofiles/pids, last access: 20 June 2023).

| Variable name | Feature | Description |
|---|---|---|
| zt | Required | Terrain height above sea level in metres |
| vegetation_type | Required | Classification of vegetation types at the land surface |
| pavement_type | Required | Classification of pavement types at the land surface |
| water_type | Required | Classification of water bodies |
| soil_type | Required | Classification of soil types, usually specified for corresponding vegetation types |
| surface_fraction | Required | Relative fraction of the respective surface type given - depending on vegetation_type, pavement_type, and water_type |
| albedo_type | Optional | Optional classification of albedo for land surface |
| water_pars | Optional | Optional parameters for water bodies, including water temperature, roughness length for momentum, emissivity, etc. |
| lad | Optional | Three-dimensional leaf area density in $\mathrm{m^2 m^{-3}}$. Required for the plant canopy model to resolve vegetation canopy |
| street_type | Optional | Optional classification of street type. Required for application of the parameterised traffic emissions and the multi-agent system |
| building_type | Optional | Classification of building types. Required for buildings in the urban surface model |
| buildings_2d | Optional | Heights of buildings relative to the underlying terrain. Required for buildings in the urban surface model |
| building_id | Optional | Building ID to identify individual building envelopes. Required for buildings in the urban surface model |
| buildings_3d | Optional | Three-dimensional building topology relative to the underlying terrain. Required for buildings in the urban surface model |

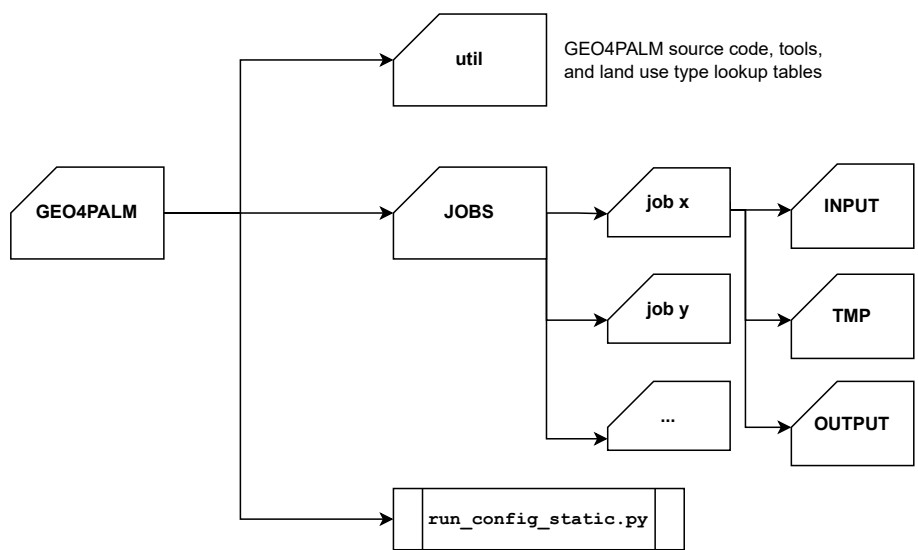

**Figure 1.** GEO4PALM file steering outline. The main executable script is `run_config_static.py`. The `util` folder contains all GEO4PALM utilities, including source code, tools and land use classification lookup table. Within the `JOBS` folder, each user-specified job is stored separately with input files in `INPUT`, temporary files in `TMP`, and output files in `OUTPUT`.

## 3 GEO4PALM framework

An early version of GEO4PALM was applied in simulations presented by Lin et al. (2021) and Lin et al. (2023). However, this version (GEOPALM v1.0) was only applicable for geospatial data for Ōtautahi / Christchurch, New Zealand. In this paper, several new features have been added to GEO4PALM v1.1. The new design of GEO4PALM v1.1 aims to 1) allow users to create static drivers for PALM simulations regardless of the geospatial data sources and 2) simplify the workflow of generating the static driver. The file steering structure of GEO4PALM is shown in Figure 1. The main source code of GEO4PALM is stored

in the `util` folder with the main executable Python script (`run_config_static.py`) located in the main directory. The `JOBS` folder allows users to create static driver files for multiple jobs (x, y, z, etc.). In each job directory, users must have an `INPUT` folder, which includes a configuration file, a land use type lookup table, and all input geospatial data files for the static driver. The `TMP` folder stores all temporary files, and all static driver files are created and stored in the `OUTPUT` folder.

The framework of GEO4PALM is shown in Figure 2. Users must at least provide a configuration file, which contains PALM

domain configuration details and the data sources of static driver input. GEO4PALM uses Python packages including xarray, rasterio, rioxarray, geopandas, and geocube (Hoyer and Hamman, 2017; Gillies et al., 2019; Jordahl et al., 2020; Snow et al., 2022a, b) to process the geo-information data. When converting geospatial data into static driver, GEO4PALM requires input files in GeoTIFF format, while no requirement is set regarding the spatial resolution and projection of the GeoTIFF files. The geospatial input data are resampled automatically based on the grid spacing values specified in the configuration

file. Users have the freedom to choose the resampling method depending on their simulation needs. For details of the avail-

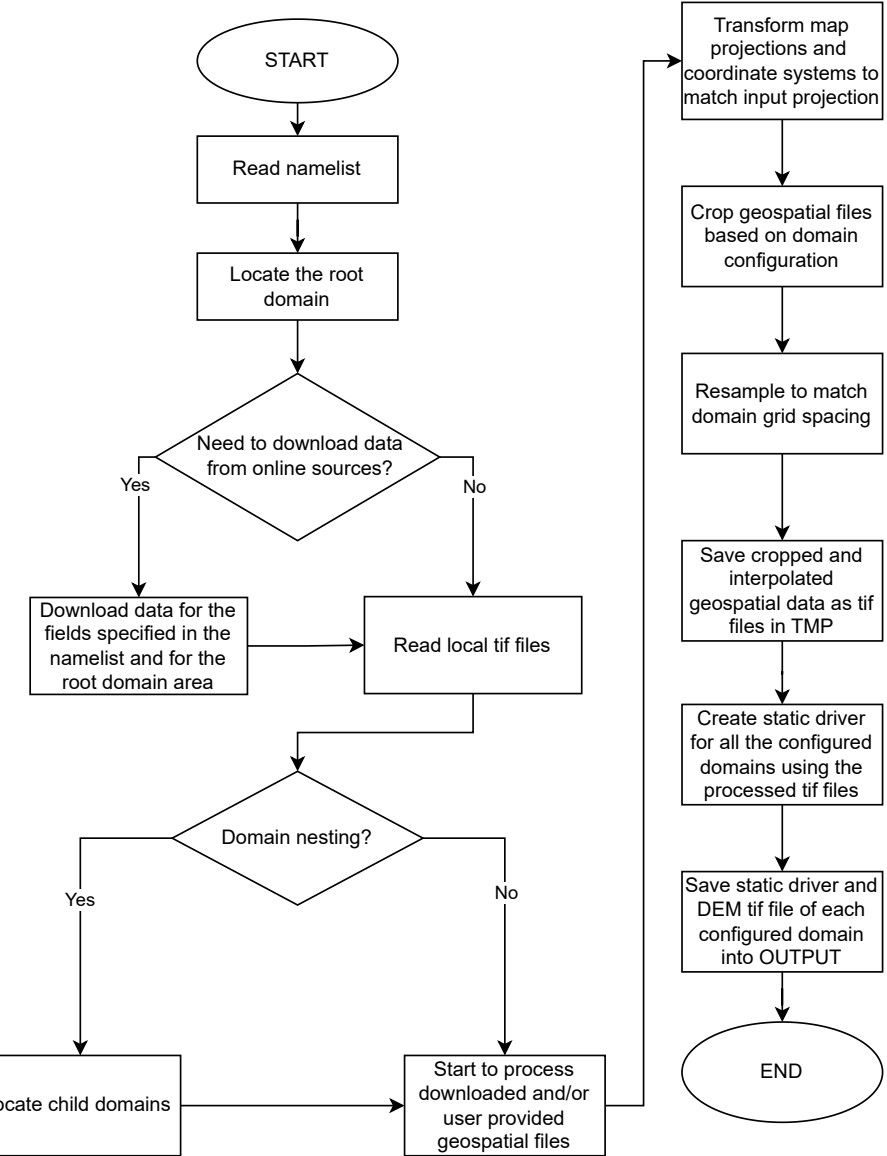

**Figure 2.** Flowchart showing the code structure of GEO4PALM.

able options, users are referred to the rasterio documentation (https://rasterio.readthedocs.io/en/stable/api/rasterio.enums.html#rasterio.enums.Resampling; last access: 23 June 2023). In addition to the GeoTIFF format, the shapefile format is one of the most common file types for geospatial data. Different from the rasterised GeoTIFF files, the shapefiles are usually vectorised with multiple layers, and each layer has its own designated name, which varies with the data set. Due to the complexity of shapefiles, it is exhaustive to include shapefiles and all the embedded layers as a direct input in GEO4PALM. Therefore, a script `shp2tif.py` is provided as a GEO4PALM pre-processing tool for users to convert shapefiles to GeoTIFF format of the desired resolution (the finest in the input configuration file by default). This script converts one layer of the shapefile at a time allowing users to choose the layer based on their applications. Table 2 explains all variables contained in the configuration file. A step-by-step guide for GEO4PALM and a sample configuration file are provided in Appendix A.

In the configuration file, users need to specify the desired geographic projection, domain configuration, and geospatial input data source for PALM simulations. If the desired projection of PALM simulations is not specified, GEO4PALM will use the nearest Universal Transverse Mercator (UTM) projection based on the latitude and longitude of the PALM domain centre given in the configuration file. To visualise PALM domain locations and to help users build the domain configuration easily, GEO4PALM is incorporated with a web-based interactive graphical user interface (GUI; Figure 3). The GUI is generated by the `palm_domain_utility.py` script. This PALM domain utility allows users to render grid boxes of PALM domains over satellite and aerial imagery obtained from Environmental Systems Research Institute (ESRI) World Imagery (https://geoviews.org/gallery/bokeh/tile_sources.html; last access: 19 June 2023) through open-source Python libraries including Panel and Geoviews (https://panel.holoviz.org/ and https://geoviews.org/; last access: 19 June 2023). The utility automatically checks and adjusts the domain configuration to avoid violations of rules for PALM domain nesting such as overlapping of domain boundaries and mismatching of grid between parent and child domains. More details for using the PALM domain GUI are described in the GEO4PALM user manual (https://github.com/dongqi-DQ/GEO4PALM; last access: 7 November 2023).

Users are not required to have a full set of input data. For each geospatial input field, users can provide their own data in GeoTIFF or shapefile format and download data from online sources when some of the data sets are not locally available. For geospatial files provided by users, one needs to specify the file name in the configuration file. If one desires to use online data, the data source should be specified. For the time being, we provide several interfaces to download data sets with global coverage. Water temperature, digital elevation model (DEM) and land use are the three mandatory elements to create a static driver. Water temperature is an important factor which can have an impact on boundary layer structure (e.g. Mahrt and Hristov, 2017), while by default PALM prescribes a water temperature of 283.0 K for all water bodies. Therefore, water temperature is required in GEO4PALM. Users can provide their own water temperature map in a GeoTIFF file or a prescribed water temperature in the configuration file for all water bodies in the simulation domains. Alternatively, users can use the online sea surface temperature (SST) data set downloaded by GEO4PALM. As SST is widely available across various global data sets, we provide an interface to download SST data to represent water temperature for water bodies in the GEO4PALM static driver. The SST data (resolution of $0.01°$) are obtained using the Earthdata Common Metadata Repository (CMR) API operated by the National Aeronautics and Space Administration (NASA) which is linked to the OPeNDAP interface (Open-source Project for a Network Data Access Protocol; https://lpdaac.usgs.gov/tools/opendap/; last access: 20 May 2023). The NASA

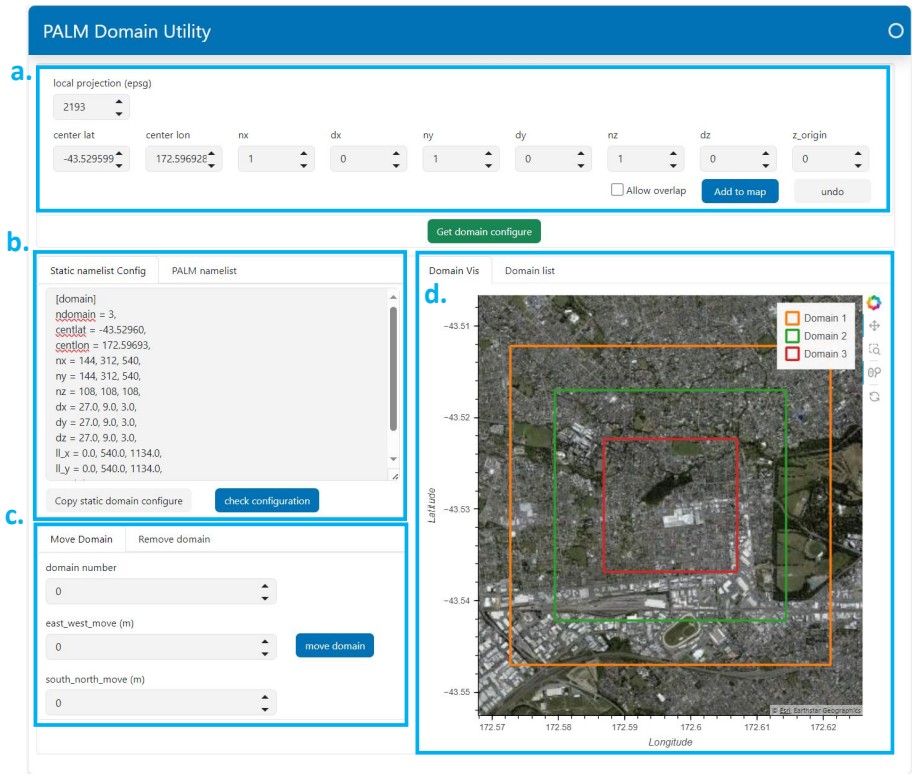

**Figure 3.** A screenshot of the web-based PALM domain utility GUI showing domain configuration for the Christchurch case described in Section 4.3. Users can input domain configuration in the subtab (a). The GUI automatically generates configuration parameters for both GEO4PALM and PALM domain configuration in the subtab (b). Users can adjust the domain locations using the subtab (c). PALM domains are drawn over the interactive satellite imagery in the subtab (d).

data sets are available globally, and users are required to register an account to use the data freely. By default, GEO4PALM downloads the version 4.1 Multiscale Ultrahigh Resolution (MUR) of a Group for High-Resolution Sea Surface Temperature (GHRSST) Level 4 analysis provided by NASA Jet Propulsion Laboratory (NASA/JPL, 2015; Chin et al., 2017). The water temperature is obtained from the nearest grid point of the SST data set to the PALM simulation domains. To download and use this data set, users must specify "`online`" in the configuration file for the variable "`water`". The date-time of the SST data should be specified using the parameter "`origin_time`" in the "`[case]`" section. If users have spatial water temperature data available for water bodies in GeoTIFF format, they can specify the data file name in the configuration file for the variable "`water`". Users are also allowed to prescribe a fixed water temperature for each simulation domain using the "`water_temperature`" parameter in the "`[settings]`" section.

The DEM (spatial resolution of 30 m) and land use classification are available to download from the Application for Extracting and Exploring Analysis Ready Samples (AρρEEARS, https://appeears.earthdatacloud.nasa.gov/; last access: 7 November 2023) operated by NASA. The DEM is the product of NASA Shuttle Radar Topography Mission 1 arc second NetCDF V003

(SRTMGL1_NC.003) acquired by spaceborne radar (Rabus et al., 2003). For the NASA data, users may provide the start and end date for the data acquisition such that the land use is representative for the simulation period. GEO4PALM source code provides a lookup table (Table B1) for the MODIS Land Cover Type Product (A$\rho\rho$EEARS product code: LC_type1), which converts the land use classification to PALM recognisable values. The MODIS Land Cover Type Product supplies global land cover maps at 500 m spatial resolution dated from 2001 (Sulla-Menashe and Friedl, 2018). More options for land use classification data sources provided by NASA refer to A$\rho\rho$EEARS online documentation (https://appeears.earthdatacloud.nasa.gov/; last access: 7 November 2023). If users desire to download and use A$\rho\rho$EEARS data sets, they must specify "`nasa`" in the configuration file for the variable "`dem`" and/or "`lu`".

In addition to the A$\rho\rho$EEARS interface, GEO4PALM incorporates the application programming interface that connects to the worldwide land cover mapping data products (WorldCover; https://esa-worldcover.org/en; last access 7 November 2023) operated by the European Space Agency (ESA). The ESA WorldCover data have spatial resolution of 10 m (Zanaga et al., 2021, 2022). Users must register a free account to obtain data from ESA. A lookup table (Table B2) for PALM readable conversion is provided in the GEO4PALM source code. For the usage of ESA data, users are required to specify "`esa`" in the configuration file for the variable "`lu`". In addition to the lookup tables for NASA and ESA data, the GEO4PALM source code provides a lookup table (Table B3) for New Zealand land cover database (LCDB) V5.0 (Landcare Research, 2020). All the lookup tables are presented in Appendix B.

All urban and plant fields can be left blank (""), in case users do not require such features in the static driver. If users desire to have other land surface features, GEO4PALM can process data for the urban surface model and plant canopy model. In PALM, urban surfaces include pavements, buildings, and streets. Although according to PIDS v1.12, the typology of streets is represented by the variable `pavement_type` and the variable `street_type` is only used for the chemistry model in PALM, GEO4PALM still includes the variable `street_type` such that the static driver can be used for simulations that require the chemistry model and/or the multi-agent system. Like the DEM and land use data, users can either provide their own GeoTIFF data, or choose to download online. If users wish to provide their own data, they must provide GeoTIFF files with building height at the building location, building ID for each building, PALM-recognisable pavement types for each pavement, and/or PALM-recognisable street types for each street, separately. Otherwise, users can specify "`osm`" for the urban variables in the configuration file to download data from OpenStreetMap (OSM; https://www.openstreetmap.org/; last access: 7 November 2023). GEO4PALM uses the OSMnx package (Boeing, 2017) to obtain data from OSM. Downloaded OSM data sets are converted to PALM-recognisable data by GEO4PALM using the conversion described by Heldens et al. (2020). Note that OSM also provides land use classification, but does not have a good spatial coverage for many regions in the world. Therefore, GEO4PALM currently does not support OSM land cover data. Users are encouraged to use the OSM land cover data set with a modified land use type conversion table like those shown in Appendix B, if their PALM simulations are conducted for regions with good spatial coverage of OSM land cover data. GEO4PALM is adaptable to any land use data provided in shapefile or GeoTIFF format.

For plant canopy, GEO4PALM currently only supports leaf area density (LAD; Lalic and Mihailovic, 2004) calculations based on vegetation height provided in the GeoTIFF files. To avoid noise from other surface geometry, GEO4PALM applies an

automatic process in which surface objects with height less than the filter (`tree_height_filter` in Table 2) are removed. With this filter, objects like cars or fences are not included as vegetation. The default value of the filter is 1.5 m, and users can adjust the value in the configuration file (`tree_height_filter` in Table 2). With high-quality data, this noise filter can be set to a desired low value ($\geq 0.0$ m) such that low objects, like grass, long grass, and bushes, can be included and represented in PALM simulations. The LAD calculation is adopted from PALM CSD (Heldens et al., 2020) and is based on the equation proposed by Lalic and Mihailovic (2004) as follows,

$$\text{LAD}(z) = \text{LAD}_m \left( \frac{h - z_m}{h - z} \right)^n \exp \left[ \left( 1 - \frac{h - z_m}{h - z} \right) n \right] \qquad [\text{m}^2 \, \text{m}^{-3}] \tag{1}$$

where $\text{LAD}_m$ is the maximum LAD, $h$ is the tree height, $z_m$ is the height where the LAD reaches $\text{LAD}_m$, and $n = 6$ when $z < z_m$ and $n = 0.5$ when $z \geq z_m$. According to Kolic (1978) and Lalic and Mihailovic (2004), the normalised value of $z_m$ ranges from 0.2 to 0.4 depending on the tree type. Currently, GEO4PALM only allows users to provide a fixed value of leaf area index (LAI; `tree_lai_max` in Table 2) and $z_m$ (`lad_max_height` in Table 2) values as input. LAI is recognised as the integration of LAD over the tree height ($h$). To derive LAD at each vertical level, GEO4PALM uses the given LAI and $z_m$ with the integral form of Equation 1 (refer to Lalic and Mihailovic (2004) for more details). GEO4PALM automatically scales the LAD based on the height of the vegetation canopy at individual grid points. This approach does not take account of spatial variation in LAD for different tree species, while it is still useful in cases where no LAD or LAI data are available. For cases in which LAD or LAI information is available, users are advised to adjust the code to directly read the spatial LAD/LAI information. However, to the best of our knowledge, no globally available vegetation height, along with plant canopy data, are currently free to obtain from any online sources. Therefore, GEO4PALM does not provide any online interface for this purpose. One possible solution to obtain vegetation height is to calculate surface objects' height using the digital surface model (DSM) and DEM. In addition to the information on ground surface altitude contained in DEM, DSM supplies the heights of all surface objects, such as buildings and trees.

Once users have provided all required information in the configuration file along with their own geospatial data where applicable, GEO4PALM downloads and/or processes the input data to create the static driver. GEO4PALM allows users to configure nested domains and use different input data for each domain. To reduce the learning curve, the domain nesting configuration is similar to PALM's nesting module, in which the nested domain location is determined by the distance of the lower left corners between the root domain and child domains. PALM's own input and output files can contain geospatial information, while the geospatial projection references sometimes may not be included accurately in a NetCDF file. Geospatial coordinates with correct geospatial projection could be important in real-world applications, especially when comparing PALM results to observations. To overcome this potential issue, instead of providing geospatial information in the NetCDF files, a GeoTIFF file with coordinate information is created by GEO4PALM along with each static driver. We recommend that GEO4PALM users use the GeoTIFF file to better reference the geospatial coordinate. For more details and examples, users are referred to the supplements, which contain the input files for the case studies presented in Section 4.

**Table 2.** Variables descriptions of GEO4PALM input configuration file.

| Variables | Descriptions | Comment |
|---|---|---|
| **case** | | |
| case_name | Name of the case, identical to the job folder name. | |
| origin_time | Date and time to start the PALM simulation in "YYYY-MM-DD HH:mm:ss +HH" format. For example, 1200 UTC on 21st June 2019 is "2019-06-21 12:00:00 +00". | Refer to origin_date_time in the PALM input configuration file for more details. |
| default_proj | The default projection (EPSG:4326) for GEO4PALM to use latitudes and longitudes to locate domains. | For most users, no changes are needed. |
| config_proj | The desired projection of the static driver. | Local projection with units in metres is recommended, for example, EPSG:2193 for New Zealand. |
| lu_table | File name of land use lookup table to convert land use classification to PALM recognisable classifications. | Should be provided in the INPUT folder. Otherwise, GEO4PALM uses the default table provided in util. |
| **settings** | | |
| water_temperature | User input water temperature values in Kelvin when no water temperature data is available. | One value for each simulation domain. |
| building_height_dummy | User input dummy height in metres for buildings where building heights are missing in the online OpenStreeMap data set or if 0.0 m is provided as building height in the input data. | One value for each simulation domain. |
| tree_height_filter | User input filter height in metres to remove small surface objects, i.e., if object height is below this value then this object is not included in the LAD estimation | One value for each simulation domain. |

| Variables | Descriptions | Comment |
|---|---|---|
| **domain** | | |
| ndomain | Maximum number of domains. | If ndomain ≥ 2, domain nesting is enabled. |
| centlat | Centre latitude of the root domain. | Not required for nested child domains. |
| centlon | Centre longitude of the root domain. | Not required for nested child domains. |
| nx | Number of grid points along the x-axis. | The actual number of grid points along the x-axis (nx). Note that this is different from the PALM configuration file (nx-1). |
| ny | Number of grid points along the y-axis. | The actual number of grid points along the y-axis (ny). Note that this is different from the PALM configuration file (ny-1). |
| nz | Number of grid points along the z-axis. | |
| dx | Grid spacing in metres along the x-axis. | |
| dy | Grid spacing in metres along the y-axis. | |
| dz | Grid spacing in metres along the z-axis. | |
| z_origin | Mean elevated terrain grid position in metres. | Default is 0.0 m. |
| ll_x | Lower left corner distance to the first domain in metres along the x-axis; only use when nesting is required. | |
| ll_y | Lower left corner distance to the first domain in metres along the y-axis; only use when nesting is required. | |
| **geotiff** | | |
| water | Input data source for water temperature. Users need to specify the input file name or data can be downloaded from OPeNDAP with the option "online". | |

| Variables | Descriptions | Comment |
|---|---|---|
| `dem` | Input data source for topographical height. Users need to specify the file name or data can be downloaded from NASA AρρRAS with the option "`nasa`". | |
| `lu` | Input data source for land use classification. Users need to specify the file name or data can be downloaded from NASA AρρRAS with the option option "`nasa`" and/or from ESA WorldCover with the option "`esa`". | A lookup table to convert land use typologies to PALM-recognisable values is required. |
| `resample_method` | Method to resample GeoTIFF files when interpolating/extrapolating to desired grid spacing. | Default is "`nearest`". |
| **If AρρRAS interface is used** | | |
| `dem_start_date` | DEM data start date in YYYY-MM-DD format. | Default is `2000-02-12` and no need to change if SRTMGL1_NC.003 data set is used. |
| `dem_end_date` | DEM data end date in YYYY-MM-DD format. | Default is `2000-02-20`, and no need to change if SRTMGL1_NC.003 data set is used. |
| `lu_start_date` | Land use data start date in YYYY-MM-DD format. | Default is `2020-10-01` for product LC_Type01. Should be changed upon users' needs. |
| `lu_end_date` | Land use data end date in YYYY-MM-DD format. | Default is `2020-10-30` for product LC_Type01 Should be changed upon users' needs. |

| Variables | Descriptions | Comment |
|---|---|---|
| **urban** | | |
| `bldh` | Input data source for building heights. Users need to specify the file name or data can be downloaded from OSM with the option "`online`". | |
| `bldid` | Input data source for the building ID. Users need to specify the file name or data can be downloaded from OSM with the option "`online`". | |
| `pavement` | Input data source for pavement types. Users need to specify the file name or data can be downloaded from OSM with the option "`online`". | |
| `street` | Input data source for street types. Users need to specify the file name or data can be downloaded from OSM with the option "`online`". | |
| **plant** | | |
| `tree_lai_max` | Input value for maximum leaf area index. | |
| `lad_max_height` | Input value for $z_m$ (range between 0.2 and 0.4) as described in Equation 1. | |
| `sfch` | Input data source for plant height above the surface. Users need to specify the file name. | Currently, no online data source/interface is available. |

## 4 Examples for real-world applications

### 4.1 Model and simulation configuration

As mentioned in Section 3, the aim of GEO4PALM is to allow users to use geospatial data for their PALM simulations seamlessly. Users may have data ready locally in GeoTIFF or shapefile format, or they can download all geospatial data freely for any location. Several other studies have used the input data described in this article and the earlier versions of GEO4PALM. For example, the static input generated by GEO4PALM was used for Christchurch International Airport to demonstrate applications of the WRF4PALM offline nesting tool (Lin et al., 2021). Lin et al. (2023) have used GEO4PALM generated static input for fog research over the city of Ōtautahi / Christchurch.

In this section, we present two examples, both using GEO4PALM and PALM CSD, to demonstrate the performance and compatibility of GEO4PALM. One example is Berlin, Germany (52.516615°N, 13.402782°E), and the other is Ōtautahi / Christchurch, New Zealand (43.529599°S, 172.596928°E). We have used the online API for both cases to obtain geospatial input. For Berlin, we have geospatial data sets prepared and stored locally containing topography, streets, buildings, water bodies, vegetation, etc., at 2 m resolution. These data have been processed by the German Space Agency (DLR), similar to those described by Khan et al. (2021) and Heldens et al. (2020). Hereafter, we refer to the local data sets for Berlin as the DLR data sets. We have used the same DLR data sets to run both PALM CSD described by Heldens et al. (2020) and GEO4PALM with map projection of EPSG:25833 (ETRS89 / UTM zone 33N) to match the projection coded in PALM CSD. At present, PALM CSD only supports ten UTM projections between zone 28N and zone 37N or a pre-processed rectangle map projection. Therefore, for Ōtautahi / Christchurch, we pre-processed all the local data to the map projection of EPSG:2193 (New Zealand Transverse Mercator) using geographic information system (GIS) software and the tools in GEO4PALM. Then, the corresponding GeoTIFF files were processed into PALM CSD compatible NetCDF files. Since PALM CSD does not provide land use classification conversion from other data sources, we first used GEO4PALM to convert our land use input data set to PALM-recognisable land use types. Then, we used GIS software with additional Python scripts to make our land use data sets compatible with PALM CSD.

Table 3 gives an overview of the simulations conducted for Berlin and Ōtautahi / Christchurch. Overall, we conducted six simulations, comprising three simulations for each of the two cities. The three simulations for Berlin are denoted as Berlin_CSD, Berlin_GEO, and Berlin_OL, which used static drivers generated by PALM CSD, GEO4PALM with the DLR data sets, and GEO4PALM with online data sets, respectively. The three simulations for Ōtautahi / Christchurch are denoted as Chch_CSD, Chch_GEO, and Chch_OL, which used static drivers generated by PALM CSD with local data sets, GEO4PALM with local data sets, and GEO4PALM with online data sets, respectively. For demonstration purposes, both examples have identical domain dimensions and grid spacing (shown in Table 4). All geospatial input data were resampled to match the grid spacing of the simulations. To demonstrate the LAD calculation in GEO4PALM, we used `tree_lai_max`$= 5.0$ and `lad_max_height`$= 0.4$, corresponding to pine trees, for both Berlin_GEO and Chch_GEO. The initialisation time was set to 0000 UTC 1 January 2021 for Ōtautahi / Christchurch and 1200 UTC 1 January for Berlin, which is midday summer time for Ōtautahi / Christchurch and around midday winter time for Berlin. Simulation time is 6 hours for both simulations. Due to

the high computation cost with fine grid spacings, here we only performed simulations using domain nesting with the finest grid spacing at 3 m, while GEO4PALM can generate a static driver with grid spacings finer than 1 m, depending on the data source.

We used the PALM model system 22.10 to conduct the simulations. For demonstration purposes, all simulations were initialised with idealised forcing. A northerly was prescribed with a wind speed of $4 \text{ m s}^{-1}$. At the initialisation, the surface potential temperature is 295.65 K with no vertical gradient in the first 2000 m from the surface and a vertical gradient of 0.3 K per 100 m for the levels above 2000 m up to the top boundary. The prognostic equation for the water vapour mixing ratio is switched off. Periodic lateral boundary conditions are used with the clear sky radiation scheme. The Radiative Transfer Model (RTM; Krč et al., 2021) and Land Surface Model (LSM; Gehrke et al., 2021) were switched on for all domains. The Urban Surface Model (USM; Resler et al., 2017), and Plant Canopy Model (PCM; Maronga et al., 2020) are only switched on for the child domains (N02 and N03).

**Table 3.** Overview of simulations for Berlin and Ōtautahi / Christchurch.

| Simulation case name | Data source | Static driver generation tools |
|---|---|---|
| **Berlin** | | |
| Berlin_CSD | DLR data set (resolution of 2 m) | PALM CSD |
| Berlin_GEO | DLR data set (resolution of 2 m) | GEO4PALM |
| Berlin_OL | Online data sets included in GEO4PALM (refer to Table 5 for details) | GEO4PALM |
| **Ōtautahi / Christchurch** | | |
| Chch_CSD | Christchurch local data sets (refer to Table 6 for details) | PALM CSD |
| Chch_GEO | Christchurch local data sets (refer to Table 6 for details) | GEO4PALM |
| Chch_OL | Online data sets included in GEO4PALM (refer to Table 6 for details) | GEO4PALM |

**Table 4.** Nested domain dimension summary. Here, x refers to the west-east coordinate, y refers to the south-north coordinate, and z refers to the vertical coordinate.

| Domain | Number of grid points (x, y, z) | Domain size (x, y, z) | Horizontal grid spacing (dx, dy) | Vertical grid spacing (dz) |
|---|---|---|---|---|
| N01 | 144*144*108 | 3888*3888*2916 | 27 m | 27 m |
| N02 | 312*312*108 | 2808*2808*972 | 9 m | 9 m |
| N03 | 540*540*108 | 1620*1620*324 | 3 m | 3 m |

## 4.2 Case study for Berlin

This section demonstrates a case study to compare the static drivers created by PALM CSD, GEO4PALM with local data, and GEO4PALM with online data. The input data sources used in GEO4PALM are listed in Table 5. For Berlin_CSD and

**Table 5.** Geospatial input data for GEO4PALM used in the Berlin case study. Refer to Khan et al. (2021) and Heldens et al. (2020) for details of data sources for Berlin_CSD.

| GEO4PALM variables | Data set | Source |
|---|---|---|
| **Berlin_GEO** | | |
| `water` | GHRSST Level 4 MUR product | OPeNDAP via Earthdata (https://lpdaac.usgs.gov/tools/opendap/; last access: 7 November 2023) |
| `dem` | DLR data sets | Refer to Khan et al. (2021) and Heldens et al. (2020). |
| `lu` | DLR data sets | Refer to Khan et al. (2021) and Heldens et al. (2020). |
| `bldh` | DLR data sets | Refer to Khan et al. (2021) and Heldens et al. (2020). |
| `bldid` | DLR data sets | Refer to Khan et al. (2021) and Heldens et al. (2020). |
| `pavement` | DLR data sets | Refer to Khan et al. (2021) and Heldens et al. (2020). |
| `street` | DLR data sets | Refer to Khan et al. (2021) and Heldens et al. (2020). |
| `sfch` | DLR data sets | Refer to Khan et al. (2021) and Heldens et al. (2020). |
| **Berlin_OL** | | |
| `water` | GHRSST Level 4 MUR product | OPeNDAP via Earthdata (https://lpdaac.usgs.gov/tools/opendap/; last access: 7 November 2023) |
| `dem` | NASA Shuttle Radar Topography Mission 1 arc second NetCDF V003 (SRTMGL1_NC.003) | A$\rho\rho$EEARS (https://appeears.earthdatacloud.nasa.gov/; last access: 7 November 2023) |
| `lu` | ESA WorldCover 10m (2020 V1) | TerraCatalogueClient (https://vitobelgium.github.io/terracatalogueclient; last access: 7 November 2023) |
| `bldh` | OSM | OpenStreetMap via osmnx (Boeing, 2017) |
| `bldid` | OSM | OpenStreetMap via osmnx (Boeing, 2017) |
| `pavement` | OSM | OpenStreetMap via osmnx (Boeing, 2017) |
| `street` | OSM | OpenStreetMap via osmnx (Boeing, 2017) |
| `sfch` | Not available | Not applicable |

Berlin_GEO, the data were processed by DLR. For details of the DLR data sets and PALM CSD, refer to Khan et al. (2021) and Heldens et al. (2020). As water temperature was not included in the DLR data set, we used GEO4PALM to obtain water temperature via the Earthdata API for Berlin_GEO. Regarding Berlin_OL, the only user input is the configuration file. GEO4PALM handled all data downloading from online sources, including the global GHRSST data, NASA 30 m DEM data (SRTMGL1_NC.003), ESA WorldCover land use classification data, and OSM urban data sets.

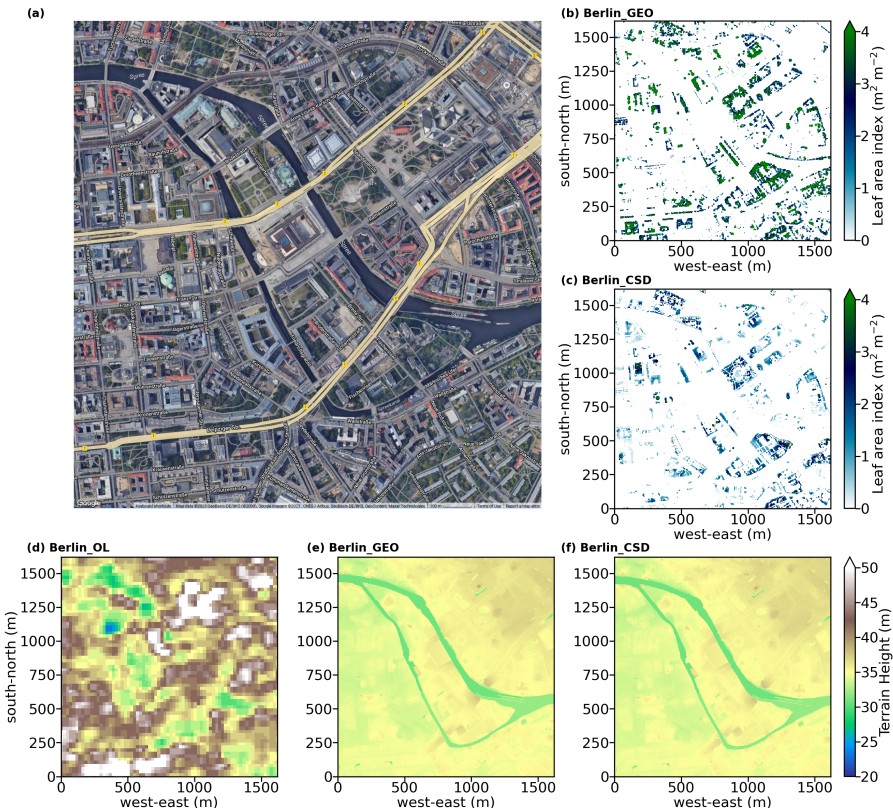

**Figure 4.** (a) Satellite images showing domain location of the nested domain N03 for the Berlin case (©Google Earth, last access: 19 May 2023). Domain centre is located near Humboldt Forum in central Berlin, Germany. The horizontal cross sections of static input data: (b-c) LAI (vertically integrated LAD), and (d-f) terrain height. Data sources refer to Table 4. No LAI data are displayed for Berlin_OL, as no online data source is available for the estimation of LAD.

The domain location and static driver data for Berlin_CSD, Berlin_GEO, and Berlin_OL are shown in Figures 4 and 5. In Berlin_OL, several buildings are included in the simulation domain. However, the OSM data (used in the Berlin_OL simu-
lation) do not contain building height for most buildings. Therefore, buildings with no height data available were assigned a height of 3.0 m for demonstration purposes. LAI, the vertical integrated LAD, shows areas with plant canopy in Figures 4b and 4c. For the Berlin_OL case, LAD was not included in the static driver input, since we do not have any DEM or DSM with spatial resolution finer than 30 m. The LAI calculated by GEO4PALM (Figure 4b) is higher than the one calculated by PALM CSD (Figure 4c). In Berlin_GEO, the vegetation patch height data were directly used by GEO4PALM to estimate LAD,
without considering vegetation type. In Berlin_CSD, however, vegetation type and the simulation season were considered. A lot of data processing is required to pre-process vegetation data for PALM CSD. For the estimation of LAD, at least the vegetation height, vegetation type, and LAI data are required. Considering the inconsistency in data sources and quality worldwide,

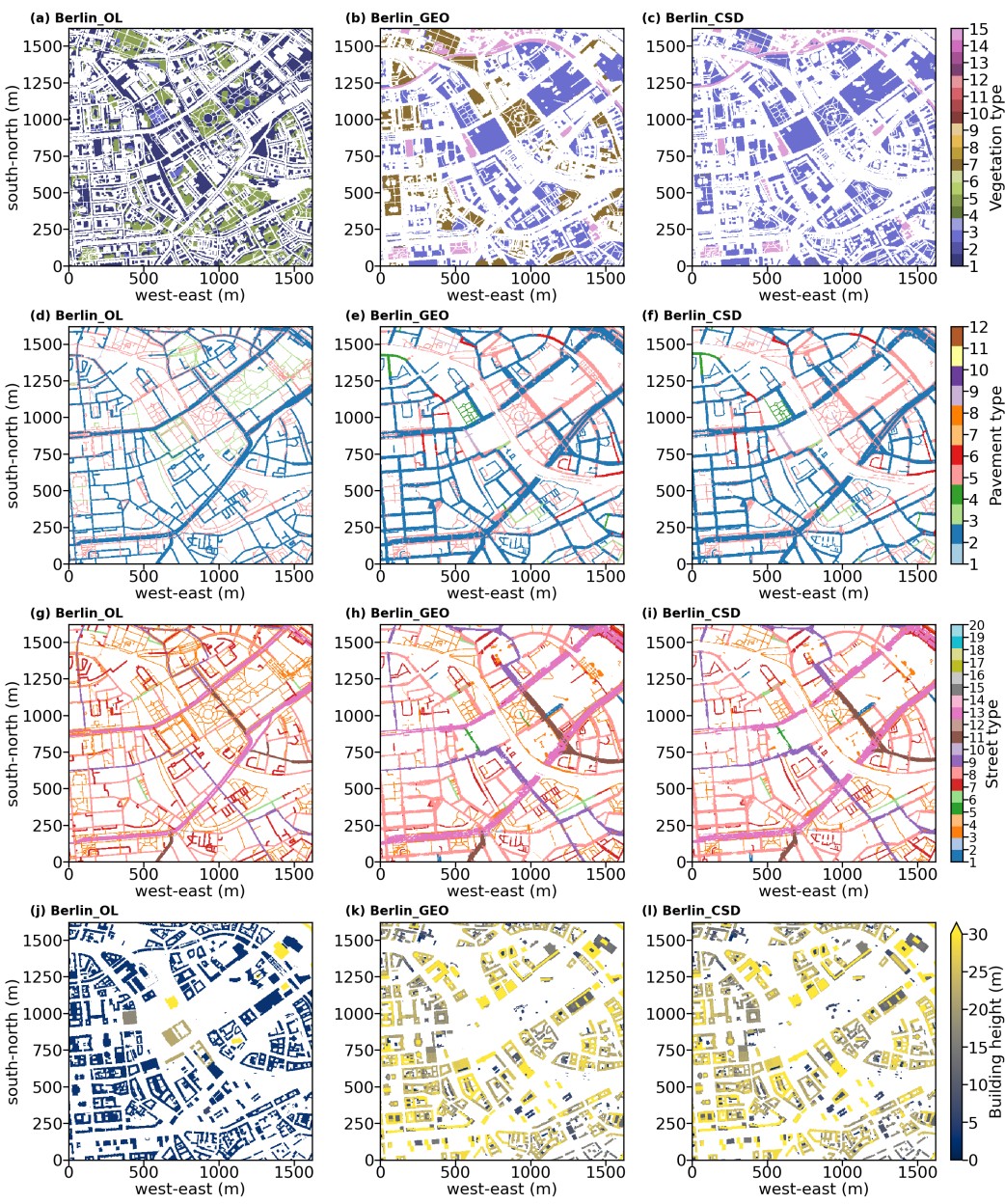

**Figure 5.** Static input data of the nested domain N03 for the Berlin case: (a-c) vegetation type, (d-f) pavement type, (g-i) street type, and (j-l) building height. Refer to the panel label for the corresponding simulation. Data sources refer to Table 5.

GEO4PALM adopts the simplified method to calculate LAD. Both GEO4PALM and PALM CSD can be modified further by users depending on their modelling needs.

Berlin_GEO and Berlin_CSD present similar features in terrain heights (Figures 4e and 4f) in which the shapes of rivers are distinguishable. However, the terrain heights in Berlin_OL (Figure 4d) do not contain many details due to the coarse resolution of its data source. The online DEM data only have a spatial resolution of 30 m. Berlin_GEO and Berlin_CSD used the same geospatial input, and hence, the topography data for the two cases are identical. It should be noted that PALM CSD offers an option to modify the terrain height of the child domains to follow the average values of their corresponding parent domain, allowing for a smoother transition of the flow at the nested child domains' lateral boundaries. After this modification, PALM CSD can be configured to adjust the topography height onto the simulation domain's vertical levels. Currently, such adjustments are not included in GEO4PALM. GEO4PALM adopts DEM directly and lets PALM itself process and convert topography into the simulation grid. These topography adjustment features are switched off for the comparison between PALM CSD and GEO4PALM presented here.

The vegetation type in Berlin_OL (Figure 5a) also has a coarser resolution compared to Berlin_GEO (Figure 5b) and Berlin_CSD (Figure 5c), because the spatial resolution for the ESA land use data is 10 m. The difference in the vegetation type classification between Berlin_OL and the other two simulations could be due to the conversion between the ESA world-cover data set and PALM. Users are referred to Table B2 for the conversion, which can be edited depending on users' needs. Comparing Berlin_GEO (Figure 5b) to Berlin_CSD (Figure 5c), one can notice that vegetation patches classified as type 7 (deciduous broadleaf trees; brown patches in Figure 5b) in Berlin_GEO are classified as type 3 (short grass; light purple in Figures 5c) in Berlin CSD. This is caused by the adjustments applied in PALM CSD. It corrects vegetation type when a vegetation height is available and is indicative of low-laying plant cover. PALM CSD also alters the vegetation type for grid points where LAD data are available, i.e., where the plant canopies are resolved. This is to avoid numerical issues when using a high roughness length with small vertical grid spacing. In addition, a tall vegetation type with high roughness length plus the resolved plant canopies could over-represent the drag of the vegetation. Subsequently, the flow reduction may be overestimated. This feature is currently not available in GEO4PALM.

The pavement type and street type presented in Berlin_OL are generally similar to Berlin_GEO and Berlin_CSD. As the widths of pavements and streets were generated automatically in Berlin_OL by GEO4PALM, some of the details, such as the width and type of each pavement and street, are of lower fidelity in Berlin_OL compared to those in Berlin_GEO and Berlin_CSD. This is similar regarding buildings. The structures and locations of buildings in Berlin_OL align with those in Berlin_GEO and Berlin_CSD. However, the DLR geospatial data do not include the Berlin Palace (as shown in the centre of Figures 4a, and 5j-5l), while this building is present in Berlin_OL (Figure 5j). This building was recently reconstructed and hence is not included in the DLR data set and subsequently not included in Berlin_GEO and Berlin_CSD.

Figures 6 and 7 illustrate the horizontal cross-sections of the simulation results. These results are hourly averages for the 2nd and 6th (i.e last) hour of the simulations. The variables displayed include 2 m potential temperature ($\theta_{2m}$), surface temperature ($T_{sfc}$), surface net radiation ($R_{net}$), and 10 m wind speed ($WS_{10m}$) and direction. To compare the simulation results between GEO4PALM and PALM CSD, the differences between Berlin_OL and Berlin_CSD, and between Berlin_GEO and

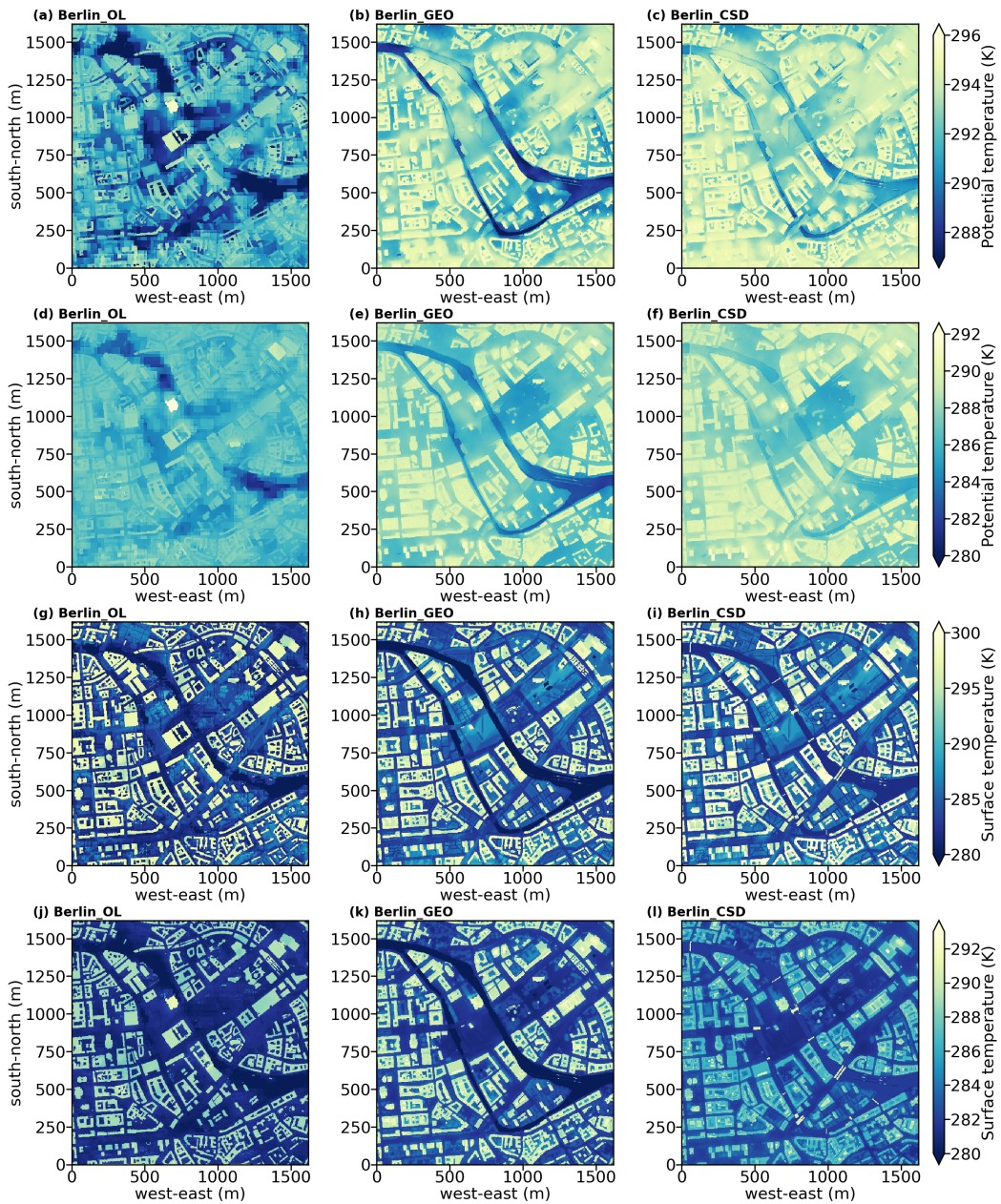

**Figure 6.** Horizontal cross-sections of simulation results for the Berlin case: (a-f) potential temperature at 2 m ($\theta_{2\mathrm{m}}$), and (g-l) surface temperature ($T_{\mathrm{sfc}}$). Panels (a-c) and panels (g-j) are averages of the 2nd hour of the simulation. Panels (d-f) and panels (j-l) are averages of the last hour of the simulations. Refer to the panel labels for the corresponding simulation.

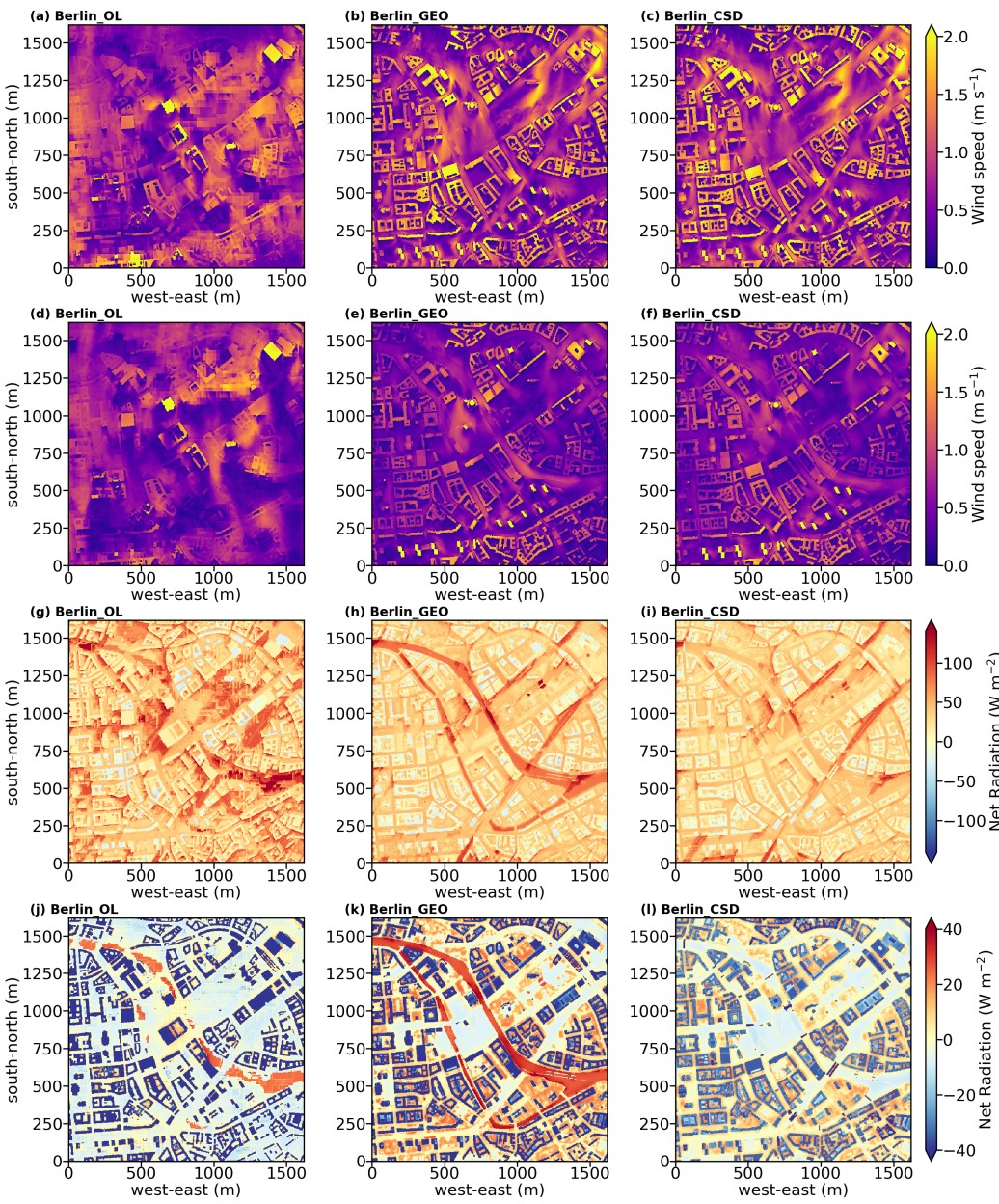

**Figure 7.** Similar to Figure 6, but for wind speed at 10 m ($WS_{10m}$) (a-f), and surface net radiation ($R_{net}$) (g-l). Refer to the panel labels for the corresponding simulation.

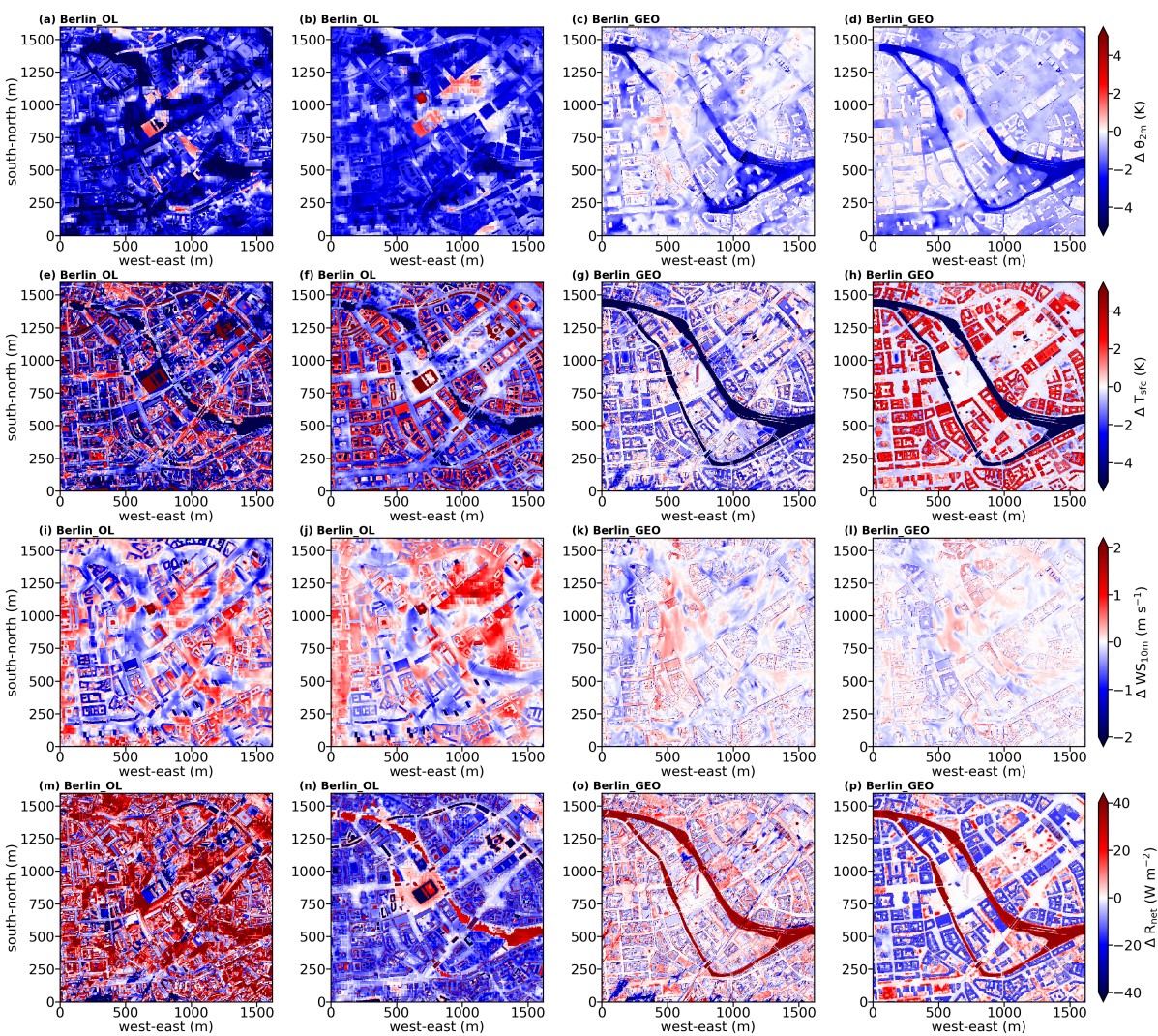

**Figure 8.** Differences of simulation results (shown in Figures 6 and 7) between GEO4PALM simulations and PALM CSD simulation. The differences are results of GEO4PALM simulations subtracted from the PALM CSD simulation: (a-d) differences in $\theta_{2m}$, (e-h) differences in $T_{sfc}$, (i-l) differences in $WS_{10m}$, and (m-p) differences in $R_{net}$. From left to right, the first and the second columns show the differences between Berlin_OL and Berlin_CSD for the 2nd and 6th hour of the simulations, respectively. The third and fourth columns show the differences between Berlin_GEO and Berlin_CSD for the 2nd and 6th hour of the simulations, respectively. Refer to the panel labels for the corresponding simulation.

Berlin_CSD are shown in Figure 8. In all simulations, $R_{net}$ and $T_{sfc}$ are strongly dependent on the land surface type and surface canopy. Comparing Berlin_OL to Berlin_GEO, the differences show the impact of the geospatial data input. The topography input data for Berlin_OL has a resolution of 30 m. The coarse topography in Berlin_OL creates box-like features, especially for $\theta_{2m}$ (see Figures 6a and 8a) and $R_{net}$ (Figure 7g) at the beginning of the simulation. Although Berlin_OL captures the building outlines in the simulation domain, the lack of accurate building height leads to an underestimation of $\theta_{2m}$ (Figure 8b) and an overestimation of $WS_{10m}$ (Figure 8j) compared to Berlin_CSD. In GEO4PALM, all buildings are configured as type 3 (residential buildings built after 2000) by default because we currently do not have a good local building type data set to use as a reference. As a result of different building-type data sets, Berlin_OL shows an overestimation of $T_{sfc}$ compared to Berlin_CSD (Figure 8f). In addition, as the land use input data have a grid spacing of 10 m for Berlin_OL, the water bodies were not presented with good fidelity in Berlin_OL, compared to Berlin_GEO and Berlin_CSD. This is reflected in the simulated $R_{net}$ in Figure 6j. As Berlin_OL uses a completely different geospatial data set compared to Berlin_CSD, the values of $R_{net}$ in Berlin_OL for non-building areas are significantly higher than those in Berlin_CSD (Figure 8m) in the 2nd simulation hour. The differences are still considerable in the last simulation hour (Figure 8n), suggesting the importance of the input data. The presence of the Berlin Palace in Berlin_OL shows a strong impact on the simulated temperature, wind, and net radiation. In more realistic applications, users should be careful about the geospatial data acquisition dates, especially in urban environments where building reconstructions frequently occur.

For a more direct comparison between GEO4PALM and PALM CSD, the results of Berlin_GEO and Berlin_CSD are presented. For the comparison in the 2nd simulation hour, Berlin_GEO coincides with more variations in $T_{sfc}$ and $R_{net}$ than in $\theta_{2m}$ and $WS_{10m}$ (Figure 8). This is because $T_{sfc}$ and $R_{net}$ are more sensitive to the differences in the input water temperature, the input building types, adjustments applied in vegetation types, and LAD estimation methods. PALM CSD allows users to input the spatial distribution of water type and water temperature. Otherwise, it uses the default water temperature of 283.0 K embedded in the code. Similar to PALM CSD, GEO4PALM allows a spatial input of water temperature. However, we do not have water temperature data available. For the Berlin case study, GEO4PALM obtained water temperature from the GHRSST data, which is 275.0 K. For comparison and demonstration purposes, we did not modify the default water temperature in PALM CSD, while users can modify the source code for their simulations. Such a difference in water temperature leads to significant contrasts in $T_{sfc}$ (Figures 8g-h) and $R_{net}$ (Figures 8o-p) between Berlin_GEO and Berlin_CSD. Over the river, $\theta_{2m}$ is also lower in Berlin_GEO. Same as Berlin_OL, only one building type is used in Berlin_GEO. This leads to higher $T_{sfc}$ in Berlin_GEO later in the simulation (Figure 8h). The differences in the building type also lead to a negative bias in the $R_{net}$ differences for building areas in Berlin_GEO.

Adjustments in the vegetation type are evident in the surface variables ($T_{sfc}$ and $R_{net}$; Figures 8g-h and 8o-p), especially at the beginning of the simulations (Figures 8g and 8o). Excluding the water bodies and buildings, the blue patches in Figure 8g coincide with the brown patches in the vegetation types of Berlin_GEO shown in Figure 5b. Without the adjustments in the vegetation type, these vegetation patches in Berlin_GEO lead to a positive bias in $R_{net}$ compared to Berlin_CSD (Figure 8o). In the last simulation hour, the cold biases in $T_{sfc}$ in Berlin_GEO over the vegetation patches are less significant (Figure 8h). Regarding $R_{net}$, the positive biases resemble the LAI patterns of Berlin_GEO shown in Figure 4b, showing the potential

impact of inaccurate LAD input. The signal of the adjusted vegetation type, however, is not clear in $WS_{10m}$ as shown in Figures 8k-l. Without the vegetation type adjustment, the wind speed is expected to be lower in Berlin_OL, while we cannot identify such an impact from Figures 8k-l. More investigation is required to determine the appropriate adjustment in vegetation types.

Overall, the static driver generated by GEO4PALM and, subsequently, the simulation fidelity of PALM is highly dependent on the input geospatial data quality. With high-quality data, GEO4PALM can create static drivers (Belin_GEO) that are comparable to PALM CSD (Berlin_CSD). Without local data, GEO4PALM can only represent the simulated environment with limited details, as presented in Berlin_OL. The lack of building height information and the coarse resolution of the online data sets may not be suitable for a realistic simulation in the urban environment. These data sets, however, could be useful for coarse simulations, for example, the parent domains of the focused simulation area.

## 4.3 Case study for Ōtautahi / Christchurch

In this section, we present a case study for Ōtautahi / Christchurch, New Zealand, to demonstrate the application of GEO4PALM when the simulation location changes. Local geospatial data sets are used to demonstrate the suitability and applicability of GEO4PALM for the case Chch_GEO. Similar to Berlin_OL, Chch_OL used all geospatial data downloaded by GEO4PALM rather than local data sets. PALM CSD was used for Ōtautahi / Christchurch using the same input as Chch_GEO, with the geospatial input files pre-processed into a specific NetCDF format. The input data for this case study are listed in Table 6. The water temperature was obtained from the GHRSST data sets for both simulations. For Chch_GEO, the DEM was obtained from Enviironment Canterbury Regional Council (2020), the land use classification was obtained from New Zealand LCDB v5.0 (Landcare Research, 2020), the building footprint, location, and building ID were derived using OSM data and New Zealand building outlines data set (Land Information New Zealand, 2020), the building height was calculated using the difference between DSM and DEM at the building locations, the pavement and street data were obtained from OSM and converted to PALM recognisable data using the conversion provided by Heldens et al. (2020), and the tree height was derived using the difference between DSM and DEM with buildings excluded.

Figures 9 and 10 show the domain location and static input data for the nested domain of 3 m grid spacing (domain N03) for Chch_OL, Chch_GEO and Chch_CSD. Riccarton Bush is located near the centre of domain N03, with sports fields to its north and Riccarton Mall (the largest building in the domain) to its south. Riccarton Bush coincides with high LAI over an area of approximately 98,000 $m^2$ as shown in Figure 9b. Again, due to the coarse resolution of the online DEM data, Chch_OL does not present good details in topography height (Figure 9d) compared to Chch_GEO (Figure 9e). However, Chch_OL does capture the decrease in topography from west to east of the domain. It should be noted that the NASA topography data may have included surface objects' heights (compare Figure 9d to Figures 9e-f), while the DEM and topography in the concept of GEO4PALM and PALM refer to the topographical height only. Users should take extra care if they use the NASA topography data. As the New Zealand land use data set (New Zealand LCDB v5.0) classifies all urban area as only one type, most area within the city of Ōtautahi / Christchurch was identified as bare soil by GEO4PALM (Figure 10b). This is different in the ESA Worldcover data set that Chch_OL shows more areas with vegetation type 5 (deciduous needleleaf trees; green in Figure 10a), which aligns with the satellite image shown in Figure 9a. Pavement type and street type are almost identical in Chch_OL

**Table 6.** Geospatial input data for GEO4PALM used in the Ōtautahi / Christchurch case study.

| GEO4PALM variables | Data set | Source |
|---|---|---|
| **Chch_GEO** | | |
| water | GHRSST Level 4 MUR product | OPeNDAP via Earthdata (https://lpdaac.usgs.gov/tools/opendap/; last access: 7 November 2023) |
| dem | Christchurch DEM with spatial resolution of 1 m | Envirionment Canterbury Regional Council (2020) |
| lu | New Zealand LCDB V5.0 | Refer to Landcare Research (2020) |
| bldh | – OSM<br>– Christchurch DEM with spatial resolution of 1 m<br>– Christchurch DSM with spatial resolution of 1 m<br>– New Zealand building outlines data set | – OpenStreetMap<br>– Envirionment Canterbury Regional Council (2020)<br>– Land Information New Zealand (2020) |
| bldid | OSM | OpenStreetMap |
| pavement | OSM | OpenStreetMap |
| street | OSM | OpenStreetMap |
| sfch | – Christchurch DEM with spatial resolution of 1 m<br>– Christchurch DSM with spatial resolution of 1 m | Envirionment Canterbury Regional Council (2020) |

| GEO4PALM variables | Data set | Source |
|---|---|---|
| **Chch_OL** | | |
| `water` | GHRSST Level 4 MUR product | OPeNDAP via Earthdata (https://lpdaac.usgs.gov/tools/opendap/; last access: 7 November 2023) |
| `dem` | NASA Shuttle Radar Topography Mission 1 arc second NetCDF V003 (SRTMGL1_NC.003) | AρρEEARS (https://appeears.earthdatacloud.nasa.gov/; last access: 7 November 2023) |
| `lu` | ESA WorldCover 10m (2020 V1) | TerraCatalogueClient (https://vitobelgium.github.io/terracatalogueclient; last access: 7 November 2023) |
| `bldh` | OSM | OpenStreetMap via osmnx (Boeing, 2017) |
| `bldid` | OSM | OpenStreetMap via osmnx (Boeing, 2017) |
| `pavement` | OSM | OpenStreetMap via osmnx (Boeing, 2017) |
| `street` | OSM | OpenStreetMap via osmnx (Boeing, 2017) |
| `sfch` | Not available | Not applicable |

(Figures 10d and 10g) and Chch_GEO (Figures 10e and 10h) as they both used OSM data. The OSM data used in Chch_OL may be more up-to-date, while Chch_GEO used local data that were processed and checked manually. In GEO4PALM, the OSM interface recognised the footpath on the Riccarton Mall rooftop parking lot as pavements. GEO4PALM removes the

buildings if they overlap with pavements, whereas PALM CSD removes the pavements when they overlap with buildings. More investigation is needed to determine a more appropriate overlap check. However, in this case, the removal of buildings could be problematic. Comparing Figure 10j to Figure 10k, Chch_OL has several buildings missing. Since Chch_GEO was sourced from both OSM and New Zealand building outlines data set (Land Information New Zealand, 2020), its building information is more comprehensive and accurate. In addition, similar to the Berlin case, OSM online data do not provide much

building height information, and hence, most buildings in Chch_OL have a dummy height of 3 m (Figure 10j).

Using the same input files, the static driver input created by GEO4PALM and PALM CSD are almost identical. The LAI input for PALM CSD is derived using the LAI calculated by GEO4PALM. In other words, the LAI shown in Figure 9c for Chch_CSD used the LAI shown in Figure 9b for Chch_GEO as an input. Since PALM CSD applies adjustments on the vegetation type (see Figures 10b-c) and checks the resolved plant canopy and its underlying vegetation types, the LAI in Chch_CSD is only

present over areas where the vegetation type is not bare soil (see Figure 9c).

Simulation results for the Christchurch case are shown in Figures 11, 12, and 13. Similar to the Berlin case, due to the coarse resolution of the input topography data, Chch_OL presents results with box-like structures, especially over the sports fields to

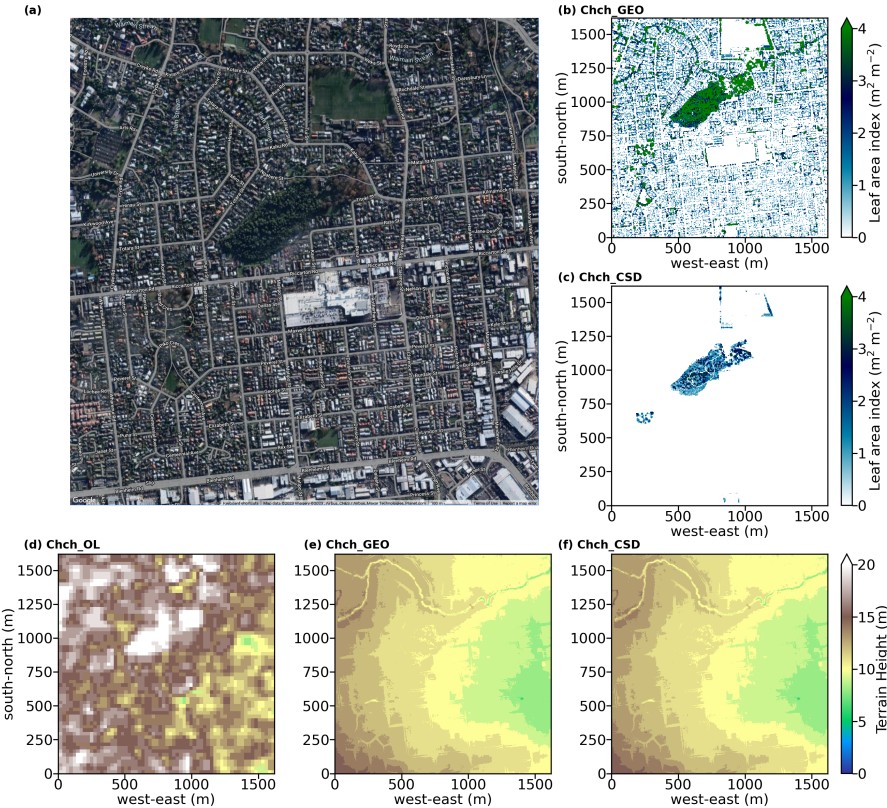

**Figure 9.** Similar to Figure 4, but for the Christchurch case. Domain centre is located near Riccarton Bush in Ōtautahi / Christchurch, New Zealand (satellite image (©Google Earth, last access: 19 May 2023). Refer to the panel label for the corresponding simulation. Data sources refer to Table 6.

the north of Riccarton Bush, where the land is exposed with no surface objects like buildings and plant canopies. The impact of the plant canopies is noticeable in Chch_GEO and Chch_OL. In contrast to Chch_OL, Chch_GEO presents lower $\theta_{2m}$ (Figures
11a-f) and $T_{sfc}$ (Figures 11g-l) over the plant canopies. Compared to Chch_CSD, the different configuration of vegetation types in Chch_OL introduces a warm bias in $\theta_{2m}$ (Figures 11a-b) and a cold bias in $T_{sfc}$ (Figures 11g and 11h) over the areas which Chch_OL recognises as vegetation rather than bare soil. The area with missing buildings in Chch_OL is colder than that in Chch_CSD (Figures 11a-b). Over the Riccarton Bush area, Chch_OL does not have a resolved plant canopy, which leads to an overestimation of $T_{sfc}$ in the area in the last hour of the simulations (13f), while most areas of the domain coincide with a cold
bias in $T_{sfc}$ compared to Chch_CSD. In addition to the missing plant canopy, the lack of building heights leads to a significant overestimation of $WS_{10m}$ in Chch_OL at the 2nd and the last hours of the simulations (Figure 13i-j). The problem of lacking plant and urban canopies is more evident for $R_{net}$ in Chch_OL. As shown in Figure 13m, the differences in $R_{net}$ are positive where the buildings and the resolved plant canopies are not present in Chch_OL. In the last simulation hour, Chch_OL gives

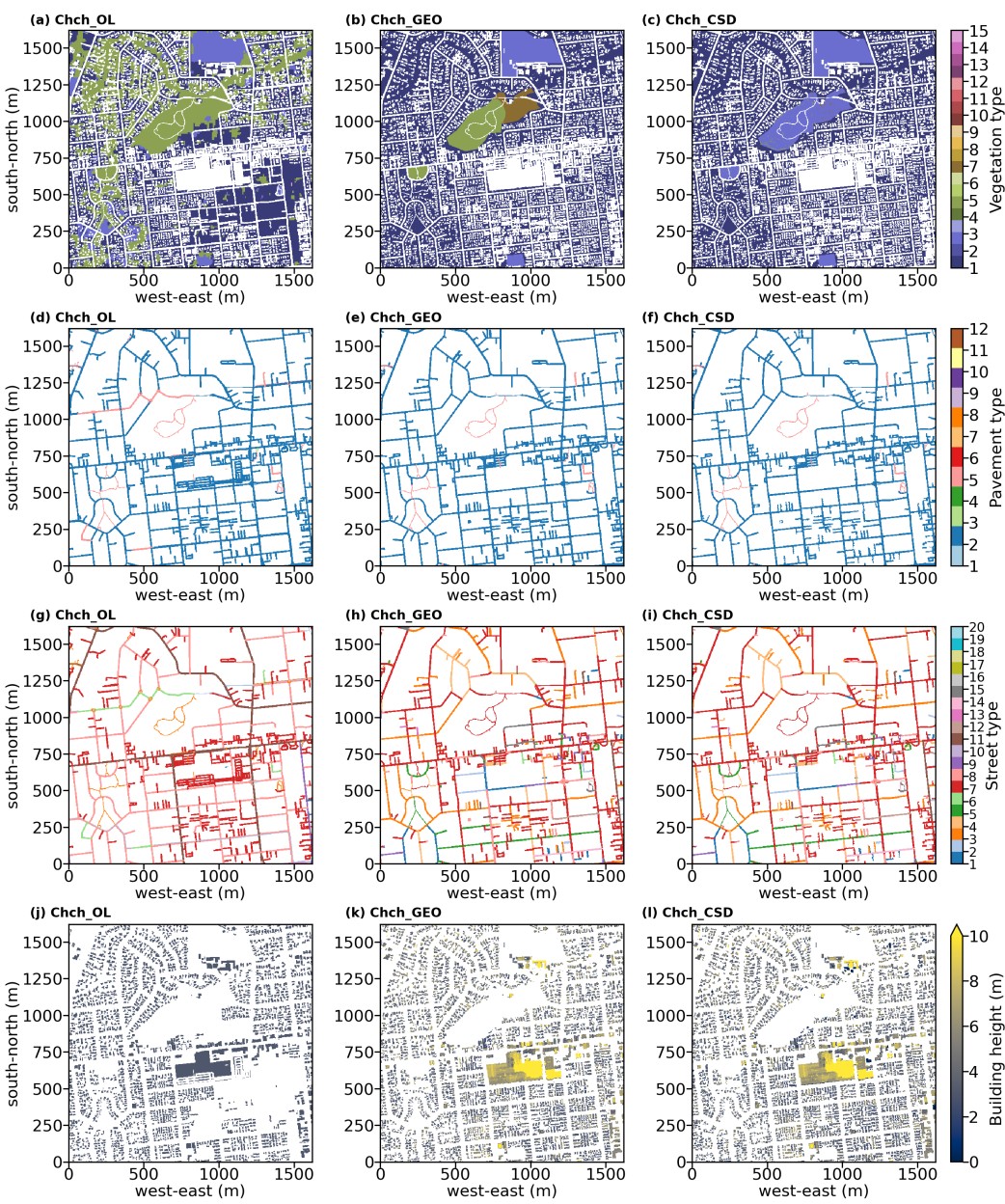

**Figure 10.** Similar to Figure 5, but for the Christchurch case. Refer to the panel label for the corresponding simulation. Data sources refer to Table 6.

a positive bias in $R_{\text{net}}$ for most areas of the domain (Figure 13n). These results, again, highlight the importance of accurate input geospatial data.

Comparing Chch_GEO to Chch_CSD, the resolved plant canopy plays a major role in the differences between the simulations. The presence of a denser plant canopy over Riccarton Bush in Chch_GEO leads to a cold bias in $\theta_{2\text{m}}$ and $T_{\text{sfc}}$ in both the 2nd and the last simulation hours (Figures 13c-d and 13g-h). In the 2nd simulation hour, with more resolved plant canopies in the simulation domain, Chch_GEO gives a higher $\theta_{2\text{m}}$ and a lower $T_{\text{sfc}}$ (Figures 13c and 13g). In the last simulation hour, the impact of plant canopies on $\theta_{2\text{m}}$ is significant that the colder areas in Chch_GEO (Figure 13d) align with the high LAI areas shown in Figure 9b. The increase in surface roughness due to the presence of the denser plant canopies in Chch_GEO also introduces a considerable decrease in wind speed compared to Chch_CSD (Figures 13)k-l). In addition, the $R_{\text{net}}$ in Chch_GEO is generally lower across the entire simulation domain than that in Chch_CSD (Figures 13)o-p). The differences between Chch_GEO and Chch_CSD agree with the assumption that, without the vegetation type adjustment and with a potentially over-estimated LAD, the simulation will create biases, especially in the simulated winds. These differences are evident in the Christchurch case because of low buildings and a higher proportion of the resolved plant canopies. In the Berlin case, the tallest building is 123.6 m with a domain-averaged building height of 23.9 m. The highest building in the Christchurch case is only 25.0 m high, and the domain-averaged building height is 4.8 m, while most of the trees in Riccarton Bush are over 12.0 m high. Therefore, in the Christchurch case, the vegetation is the main surface forcing, and such forcing could alter the simulation results significantly.

## 5    Conclusions

This study presents GEO4PALM, a utility written in Python that generates PALM static driver input for the PALM model system. With GEO4PALM, PALM users now have the freedom to create static drivers with any geospatial data they have had and/or the online data provided. Two application examples were given to demonstrate the applicability of GEO4PALM. The results show that GEO4PALM is a useful tool to realise near-surface microscale structures. When the same geospatial input data are used, the GEO4PALM static drivers can present simulation results comparable to the PALM CSD static drivers. There are differences between the results generated by the two tools. However, without validations against observational data, we cannot verify which tool's approach is more appropriate. The optimal simulation setup needs to be investigated in future simulations and experiments along with observational data. To the best of our knowledge, current static driver generation tools in the PALM community (PALM CSD, palmpy, and PALM-4U GUI) either heavily rely on users to pre-process geospatial data, or have not been applied to many regions in the world, for example, New Zealand. GEO4PALM simplifies data acquisition by automating and standardising data pre-processing and conversion. The online data interfaces, automatic projection conversion, and translation of land cover classes make GEO4PALM a widely applicable tool. These features, however, are not yet implemented in PALM CSD. In addition to the static driver generation code, GEO4PALM provides a GUI for users to visualise and configure the simulation domains easily.

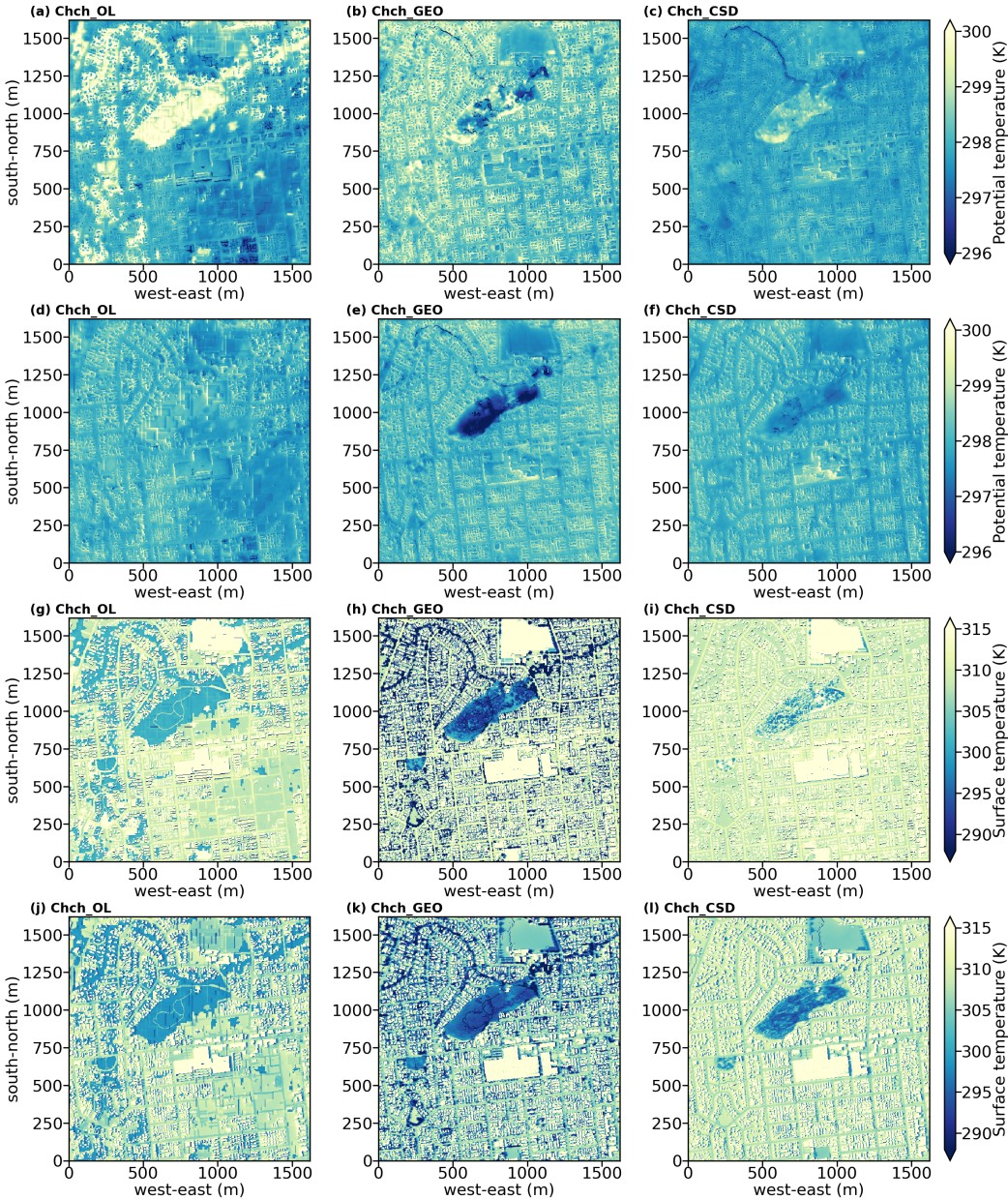

**Figure 11.** Similar to Figure 6, but for the Christchurch case. Refer to the panel label for the corresponding simulation.

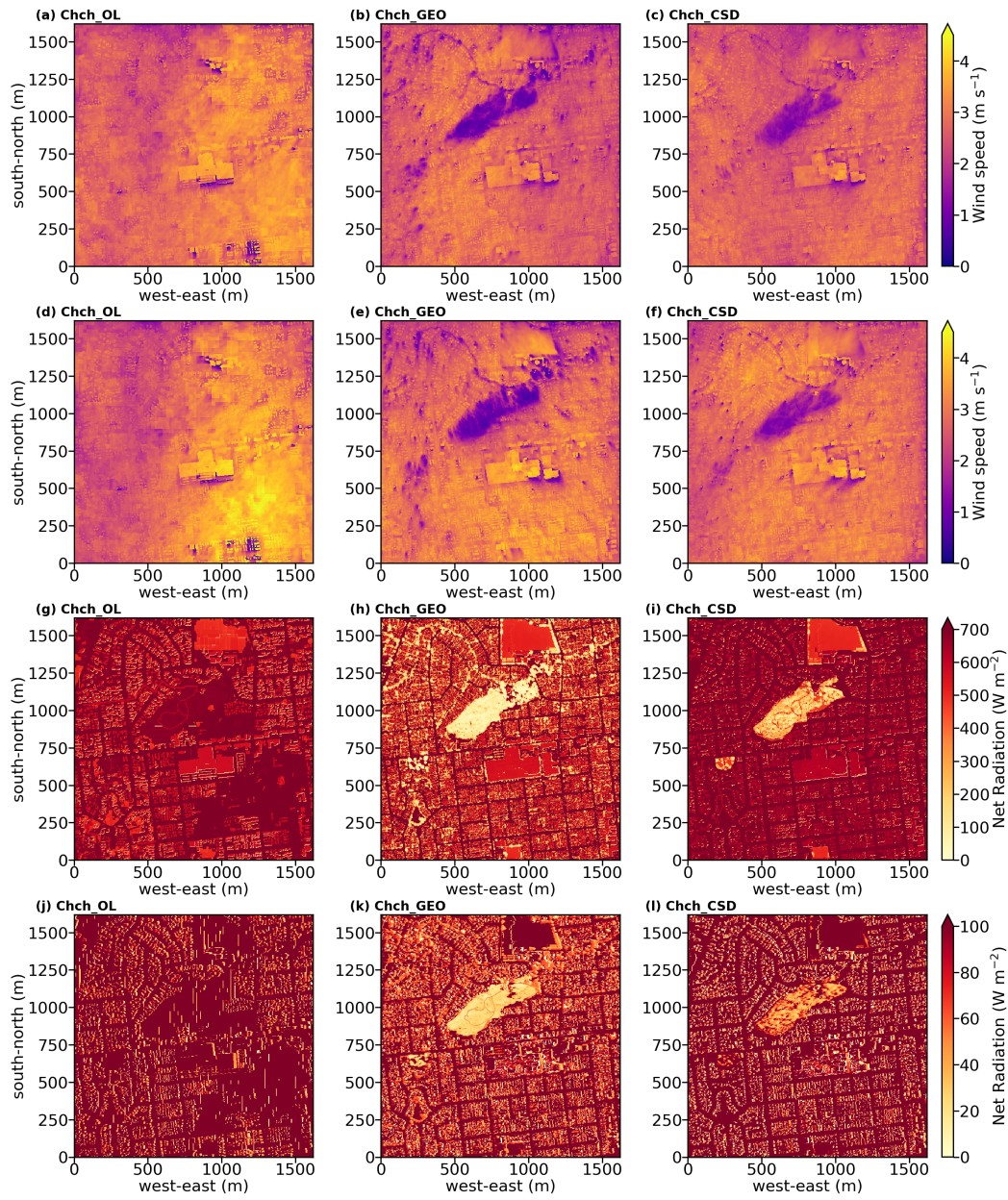

**Figure 12.** Similar to Figure 7, but for the Christchurch case. Refer to the panel label for the corresponding simulation.

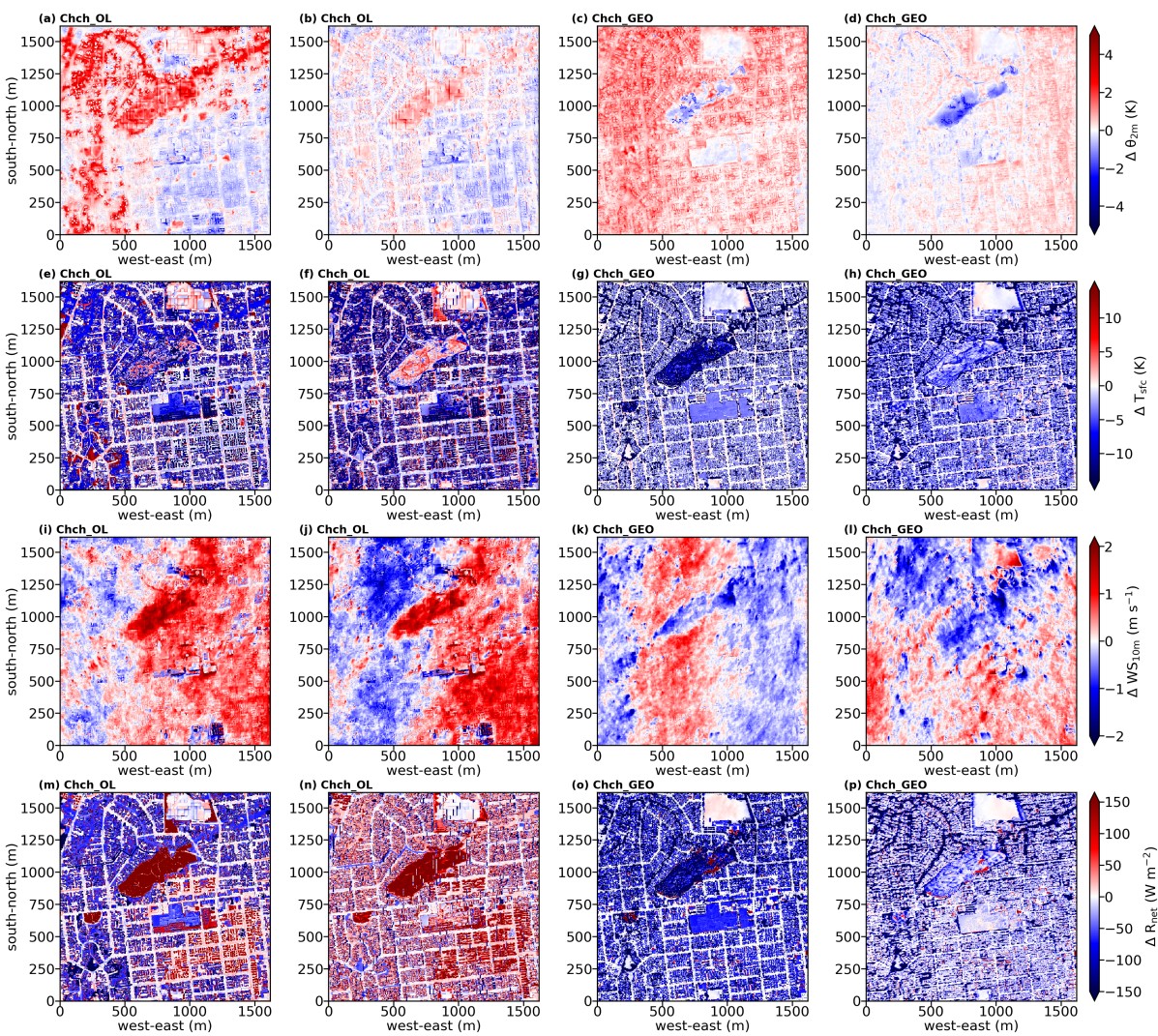

**Figure 13.** Similar to Figure 8, but for the Christchurch case. Refer to the panel label for the corresponding simulation.

**Limitations**

GEO4PALM is developed as a free, open-source, and community-driven tool and is distributed on GitHub (https://github. com/dongqi-DQ/GEO4PALM; last access: 7 November 2023). As discussed, GEO4PALM does not cover all the variables in the static driver of the PALM model system. The fidelity and features of static driver input depend on the input geospatial
data quality and availability, which have been a particular challenge when conducting microscale simulations. GEO4PALM provides several interfaces for users to download global geospatial data sets, which include the basic features of PALM static input, such as topography and land use typology. However, as shown in the case studies, these online data sets are of coarse resolution and are not suitable for high-resolution urban applications at the meter scale. These online data sets could be handy for quickly conducting simulations over a large area with coarse resolutions. The building height and plant canopy information
are currently missing in the online data sets covered by GEO4PALM. The LAD estimation method in GEO4PALM may lead to biases in simulations for highly vegetated areas. High-resolution building type and water temperature data sets are currently not included in GEO4PALM, while these data sets are not publicly available. In addition, GEO4PALM does not apply any adjustments to vegetation types and/or topography. Whether the adjustments would improve the simulations requires further investigation, which is out of the scope of this paper, but should be considered in future developments of GEO4PALM.

**Outlook**

Many regions worldwide do not have high-resolution geospatial data to realise simulations with high fidelity. High-resolution building height data are important for urban applications, but they are not widely available, making simulations over human settlement difficult. While building height information could sometimes be obtained from the local authorities in large cities, the availability of such data sets might vary depending on the regions of interest. Additionally, for research purposes, there might be
differences in accessibility and cost, particularly in smaller or less developed areas. Fortunately, with increasing efforts towards research at the microscale, especially urban climate research, new data sets have been developed for microscale simulations. For example, high-resolution geospatial data can be obtained from Geoscape (https://geoscape.com.au/; last access: 7 November 2023) for applications in Australia. Microsoft provides AI-assisted building footprint mapping (https://www.microsoft.com/ en-us/maps/building-footprints; last access: 7 November 2023). Using the deep neural network, the GLObal Building heights
for Urban Studies (GLOBUS) gives a novel Level of Detail-1 (LoD-1) building data set (Kamath et al., 2022). Esch et al. (2022) presented World Settlement Footprint 3D, which provides three-dimensional morphology and density of buildings worldwide. Some of these data sets are not freely available or need to be acquired based on individual requests, while GEO4PALM accepts all geospatial data in GeoTIFF format. Once users have obtained the data sets they desire, GEO4PALM is able to process such data for PALM simulations.

Another common challenge in land use data sets is that many land use classification data sets only classify urban areas into a limited number of typologies. This could lead to a loss of fidelity. Lipson et al. (2022) has described a data transformation method to make the urban land use data more descriptive. This may potentially improve the quality of land use classification data sets, for example, for applications in New Zealand, where only one type of land use was classified for urban areas.

GEO4PALM currently only accepts vegetation heights as input for plant canopy due to a lack of geospatial data. In the future, we aim to improve this feature based on the PALM CSD tool (Heldens et al., 2020) and to include more data sources if available. While PALM CSD applies various checks and modifications of the input geospatial data, e.g., topography and vegetation types, these features are not yet implemented in GEO4PALM. As shown in the Christchurch case study, the vegetation type adjustments could substantially impact the simulation results. Furthermore, this paper only compared GEO4PALM to PALM CSD, while palmpy and PALM-4U GUI are the other tools that could have more advanced features. Hence, in addition to the investigation and development of vegetation type adjustment, future work may also include the comparison between the four tools for better improvements on GEO4PALM.

GEO4PALM accepts any geospatial data sets as input and is easily adaptive to new data downloading interfaces. With the development of geospatial data sets towards better spatial coverage and data quality, GEO4PALM can be improved and extended. All PALM users are encouraged to provide feedback and report bugs and issues via the issue system provided by GitHub. Any optimisation, modification, or contribution to the code is welcome and much appreciated.

*Code availability.* The PALM model system is a free and open-source numerical model distributed on GitLab (https://gitlab.palm-model.org/releases/palm_model_system/-/releases; last access: 7 November 2023) under the GNU General Public License v3.0. The exact PALM model source code used for this study is release 22.10 (https://gitlab.palm-model.org/releases/palm_model_system/-/releases/v22.10; last access: 7 November 2023). GEO4PALM code is freely available at https://doi.org/10.5281/zenodo.8062321 and https://github.com/dongqi-DQ/GEO4PALM (last access: 24 November 2023) under the GNU General Public License v3.0. Details of Python packages and the environment used for GEO4PALM are given on the GitHub repository. PALM CSD code is included in the PALM source code with technical information available at https://palm.muk.uni-hannover.de/trac/wiki/doc/app/iofiles/pids/palm_csd (last access: 7 November 2023).

*Data availability.* All PALM input files for the Berlin and the Christchurch cases described in Section 4 are available in the supplement. The GEO4PALM and PALM CSD configuration files for both the Berlin and Christchurch cases are included in the supplement. Geospatial data availability for the application examples refer to the main text. Other data sets can be provided upon request.

## Appendix A: GEO4PALM step-by-step guide

A more detailed user manual is available at https://github.com/dongqi-DQ/GEO4PALM (last access: 7 November 2023).

### A1 Step 1: prepare configuration file

The configuration file should be provided in `./JOBS/case_name/` folder as follows:

```
[case]
case_name        -  name of the case
origin_time      -  date and time at model start*
```

```
     default_proj        –   default is EPSG:4326. This projection uses lat/lon to
                             locate domain. This may not be changed.
config_proj         –   projection of input tif files. GEO4PALM will automatically assign
                             the UTM zone if not provided.
                             We recommend users use local projection with units in metre,
                             e.g. for New Zealand users, EPSG:2193 is a recommended choice.
     lu_table            –   land use look up table to convert land use classification
to PALM recognisable

     [settings]
     water_temperature     –   user input water temperature values when
                               no water temperature data is available
building_height_dummy –   user input dummy height for buildings
                               where building heights are missing in the
                               OSM data set or if building heights are
                               0.0 m in the input data
     tree_height_filter    –   user input to filter small objects, i.e., if
object height is smaller than this value
                               then this object is not included in the LAD estimation

     [domain]
     ndomain             –   maximum number of domains, when >=2, domain nesting is enabled
centlat, centlon    –   centre latitude and longitude of the first domain. Note this is
                             not required for nested domains
     nx                  –   number of grid points along the x-axis
     ny                  –   number of grid points along the y-axis
     nz                  –   number of grid points along the z-axis
dx                  –   grid spacing in metres along the x-axis
     dy                  –   grid spacing in metres along the y-axis
     dz                  –   grid spacing in metres along the z-axis
     z_origin            –   elevated terrain mean grid position in metres
                             (leave as 0.0 if unknown)
ll_x                –   lower left corner distance to the first domain in
                              metres along x-axis
     ll_y                –   lower left corner distance to the first domain in
```

```
                         metres along y-axis

[geotif]            -  required input from the user; can be provided by users
                            in the INPUT folder or "online"
     water               -   input for water temperature
     dem                 -  digital elevation model input for topography
     lu                  -  land use classification
resample_method     -  method to resample GeoTIFF files for interpolation/extrapolation

     # if NASA API is used format in YYYY-MM-DD
     # SST date should be the same as the orignin_time

## No need to change start/end dates for NASA SRTMGL1_NC.003
     dem_start_date = '2000-02-12',
     dem_end_date = '2000-02-20',
     ## start/end dates for land use data set
     lu_start_date = '2020-10-01',
lu_end_date = '2020-10-30',

     [urban]             -  input for urban canopy model; can leave as "" if this
                            feature is not included in the simulations, or provided by
                            users; or online from OSM
bldh                -  input for building height
     bldid               -  input for building ID
     pavement            -  input for pavement type
     street              -  input for street type

[plant]             -  input for plant canopy model; can leave as "" if this
                            feature is not included in the simulations,
                            or provided by users
     tree_lai_max        -  input value for maximum leaf area index (LAI)
     lad_max_height      -  input value for the height where the leaf area density (LAD)
reaches LADm
     sfch                -  input for plant height; this is for leave area density (LAD)
```

## A2   Step 2: provide input files where applicable

Move all input geospatial data to `./JOBS/case_name/INPUT` folder.

## A3   Step 3: run the main script

`python run_config_static.py case_name`

## Appendix B:  Land use type lookup table

**Table B1.** Lookup table to convert NASA LC_Type01 classes to PALM vegetation type, pavement type, building type, water type, and soil type.

| Class code | Class name | vegetation type | pavement type | building type | water type | soil type |
|:---:|:---:|:---:|:---:|:---:|:---:|:---:|
| 1 | Evergreen Needleleaf Forests | 4 | | | | 5 |
| 2 | Evergreen Broadleaf Forests | 6 | | | | 5 |
| 3 | Deciduous Needleleaf Forests | 5 | | | | 5 |
| 4 | Deciduous Broadleaf Forests | 7 | | | | 5 |
| 5 | Mixed Forests | 17 | | | | 5 |
| 6 | Closed Shrublands | 15 | | | | 5 |
| 7 | Open Shrublands | 16 | | | | 5 |
| 8 | Woody Savannas | 17 | | | | 4 |
| 9 | Savannas | 17 | | | | 4 |
| 10 | Grasslands | 8 | | | | 3 |
| 11 | Permanent Wetlands | 14 | | | | 6 |
| 12 | Croplands | 11 | | | | 4 |
| 13 | Urban and Built-up Lands | | | 3 | | |
| 14 | Cropland/Natural Vegetation Mosaics | 2 | | | | 5 |
| 15 | Permanent Snow and Ice | 13 | | | | 1 |
| 16 | Barren | 12 | | | | 1 |
| 17 | Water Bodies | | | | 2 | |
| 255 | Unclassified | | | | 2 | |
| -9999 | Not a number or no data | | | | 2 | |

**Table B2.** Lookup table to convert ESA WorldCover classes to PALM vegetation type, pavement type, building type, water type, and soil type.

| Class code | Class name | vegetation type | pavement type | building type | water type | soil type |
|---|---|---|---|---|---|---|
| 10 | Tree cover | 5 | | | | 4 |
| 20 | Shrubland | 15 | | | | 4 |
| 30 | Grassland | 3 | | | | 3 |
| 40 | Cropland | 2 | | | | 3 |
| 50 | Built-up | | | 3 | | |
| 60 | Bare / sparse vegetation | 1 | | | | 1 |
| 70 | Snow and ice | 13 | | | | 1 |
| 80 | Permanent water bodies | | | | 2 | |
| 90 | Herbaceous wetland | 14 | | | | 6 |
| 95 | Mangroves | 6 | | | | 6 |
| 100 | Moss and lichen | 3 | | | | 6 |
| -9999 | Not a number or no data | | | | 2 | |
| 255 | Not a number or no data | | | | 2 | |

**Table B3.** Lookup table to convert NZLCDB v5.0 classes to PALM vegetation type, pavement type, building type, water type, and soil type.

| Class code | Class name | vegetation type | pavement type | building type | water type | soil type |
|---|---|---|---|---|---|---|
| 1 | Built-up Area (settlement) | | | 3 | | |
| 2 | Urban Parkland/Open Space | 3 | | | | 3 |
| 5 | Transport Infrastructure | | 3 | | | 1 |
| 6 | Surface Mine or Dump | 1 | | | | 1 |
| 10 | Sand or Gravel | 1 | | | | 1 |
| 11 | River and Lakeshore Gravel and Rock | | | | 2 | |
| 12 | Landslide | 1 | | | | 1 |
| 13 | Alpine Gravel and Rock | 1 | | | | 1 |
| 14 | Permanent Snow and Ice | 1 | | | | 1 |
| 15 | Alpine Grass/Herbfield | 3 | | | | 6 |
| 16 | Gravel or Rock | 1 | | | | 1 |
| 20 | Lake or Pond | | | | 1 | |
| 21 | River | | | | 2 | |
| 22 | Estuarine Open Water | | | | 2 | |
| 30 | Short-rotation Cropland | 2 | | | | 6 |
| 33 | Orchards, Vineyards or Other Perennial Crops | 11 | | | | 6 |
| 40 | High Producing Exotic Grassland | 3 | | | | 6 |

| Class code | Class name | vegetation type | pavement type | building type | water type | soil type |
|---|---|---|---|---|---|---|
| 41 | Low Producing Grassland | 3 | | | | 6 |
| 43 | Tall Tussock Grassland | 8 | | | | 6 |
| 44 | Depleted Grassland | 3 | | | | 1 |
| 45 | Herbaceous Freshwater Vegetation | 14 | | | | 6 |
| 46 | Herbaceous Saline Vegetation | 16 | | | | 6 |
| 47 | Flaxland | 8 | | | | 6 |
| 50 | Fernland | 3 | | | | 6 |
| 51 | Gorse and/or Broom | 8 | | | | 6 |
| 52 | Manuka and/or Kanuka | 8 | | | | 6 |
| 54 | Broadleaved Indigenous Hardwoods | 7 | | | | 6 |
| 55 | Sub Alpine Shrubland | 15 | | | | 6 |
| 56 | Mixed Exotic Shrubland | 15 | | | | 6 |
| 58 | Matagouri or Grey Scrub | 8 | | | | 6 |
| 60 | Minor Shelterbelts | 18 | | | | 6 |
| 61 | Major Shelterbelts | 18 | | | | 6 |
| 64 | Forest - Harvested | 17 | | | | 6 |
| 68 | Deciduous Hardwoods | 5 | | | | 6 |
| 69 | Indigenous Forest | 6 | | | | 6 |
| 70 | Mangrove | 14 | | | | 6 |
| 71 | Exotic Forest | 7 | | | | 6 |
| -9999 | Not a number or no data | | | | 2 | |

*Author contributions.* DL was responsible for the data acquisition, conceptualisation of the GEO4PALM tool, initial and major development of GEO4PALM, GEO4PALM code distribution and documentation, conducting PALM simulations, formal analysis, and visualisation. JZ contributed to the conceptualisation of GEO4PALM v1.1, major GEO4PALM development, GEO4PALM documentation, and developed the PALM domain utility GUI. BK provided the DLR data sets and contributed to the conceptualisation of case studies. DL wrote the manuscript with contributions from JZ, BK, MK, and LER. DL, JZ, BK, MK, and LER reviewed the manuscript.

*Competing interests.* The authors declare that they have no conflict of interest.

*Acknowledgements.* The contributions of Dongqi Lin and Jiawei Zhang were funded by the New Zealand Ministry of Business, Innovation and Employment (MBIE) project "Extreme wildfire: Our new reality – are we ready?" (Grant No. C04X2103). Jiawei Zhang also
received support from the MBIE project "Vive la résistance - achieving long-term success in managing wilding conifer invasions" (Grant No.CO4X2102). Marwan Katurji was supported by the MBIE extreme wildfire project (Grant No. C04X2103), and the Royal Society of New Zealand (Grant No. RDF-UOC1701). The contribution of Basit Khan was supported by the MOSAIK-2 project, which is funded by the German Federal Ministry 440 of Education and Research (BMBF) (Grant No. 01LP1911H) and Arabian Center for Climate and Environmental Sciences (ACCESS), through the New York University Abu Dhabi (NYUAD) Research Institute Grant CG009. Laura Revell
appreciates support by the Rutherford Discovery Fellowships from New Zealand Government funding, administered by the Royal Society Te Apārangi. We performed PALM simulations presented in this study on New Zealand eScience Infrastructure (NeSI) high-performance computing facilities. GEO4PALM development was conducted on the School of Earth and Environment (SEE) computing cluster and the University of Canterbury high-performance research computing cluster (RCC). The early development of GEO4PALM was inspired by the WRF2PALM code (now replaced by the WRF4PALM toolkit) developed by Ricardo Faria from the Oceanic Observatory of Madeira. We
would like to acknowledge Dr Alena Malyarenko from the University of Canterbury for internal proofreading of the manuscript. We would like to thank all the open-source Python package developers. Without their efforts, GEO4PALM cannot be built.

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
