# Peer review of "GEO4PALM v1.1: an open-source geospatial data processing toolkit for the PALM model system"

_Geoscientific Model Development, 2023_

## Author Response (AR1)

**Author's response for gmd-2023-150:**

**GEO4PALM v1.1: an open-source geospatial data processing toolkit for the PALM model system**

The authors thank the reviewers for their time, consideration, and insightful feedback. We have included discussions on the other PALM static driver tools that were brought to our awareness by the reviewers. The main suggestions from both reviewers are on the comparison of PALM results between GEO4PALM and the PALM create static driver (CSD) tool (hereafter PALM CSD). Therefore, for a better comparison between the tools, we have conducted a new simulation for the Christchurch case using PALM CSD and rerun the Berlin case using PALM CSD with the topography adjustments switched off, as suggested by Reviewer #2. We have added more discussions on the features, limitations, and outlook of GEO4PALM.

The revised manuscript has the addition of Māori city names in accordance with New Zealand government regulations.

The reviewer's comments have been listed below in black and responded to individually in blue. Revised sentences are in red.

**Reply to Anonymous Referee #1**
**Major issues**

One of the main features of GEO4PALM compared to palm_csd is the support of the automatic download and usage of freely available online data. Here, I miss a thorough discussion of the applicability of this data in the context of building-resolving LES simulations with grid-spacings down to 1m. For example, the authors mention that the used sea surface temperate data has a resolution of 0.01°. Is this resolution sufficient to derive water temperatures for rivers and lakes? Even with finer remote-sensing-based products, the clear identification of smaller water bodies is difficult. Another example is the building information, which can be downloaded from OpenStreetMap (OSM). For both exemplary domains, the building height is missing for most of the buildings and a default height of 3m is used. The authors state that this "represent(s) the simulated environment with reasonable details" (L277). I do not agree; for a building-resolving model, the building height is a crucial part of the input data.

We agree with the reviewer that the online data sets provided by GEO4PALM do not have good resolution for building-resolving large eddy simulations (LES), while these data sets could be useful for simulations over a large area with a coarse resolution, for example, the parent domains of the focused simulation region. To clarify this, we have added discussions as follows:

In Section 4.1 Case study for Berlin:

The lack of building height information and the coarse resolution of the online data sets may not be suitable for a realistic simulation in the urban environment. These data sets, however, could be useful for coarse simulations, for example, the parent domains of the focused simulation area.

In Section 5 Conclusions:

GEO4PALM provides several interfaces for users to download global geospatial data sets, which include the basic features of PALM static input, such as topography and land use typology. However, as shown in the case studies, these online data sets are of coarse resolution and are not suitable for high-resolution urban applications at the meter scale. These online data sets could be handy for quickly conducting simulations over a large area with coarse resolutions.

We agree with the reviewer that sea surface temperate (SST) data has a coarse resolution and should not be used to map the locations of water bodies. GEO4PALM obtains the water temperature from the nearest grid point of the SST data set. The SST may not be suitable for simulations with fine resolution of, for example, 1 m. It was not explained explicitly in the original manuscript that users can input their own calibrated water temperature files. In addition, we have revised the code and added an option for users to prescribe a fixed water temperature for all water bodies in the simulation domain in case users have a better estimation of the water temperature. These have been clarified and revised as follows:

Users can provide their own water temperature map in a GeoTIFF file or a prescribed water temperature in the configuration file for all water bodies in the simulation domains. Alternatively, users can utilise the online sea surface temperature (SST) data set downloaded by GEO4PALM.

By default, GEO4PALM downloads the version 4.1 Multiscale Ultrahigh Resolution (MUR) of a Group for High-Resolution Sea Surface Temperature (GHRSST) Level 4 analysis provided by NASA Jet Propulsion Laboratory (NASA/JPL, 2015). The water temperature is obtained from the nearest grid point of the SST data set to the PALM simulation domains. To download and use this data set, users must specify "online" in the configuration file for the variable "water". The datetime of the SST data should be specified using the parameter "origin_time" in the "[case]" section. If users have spatial water temperature data available for water bodies in GeoTIFF format, they can specify the data file name in the configuration file for the variable "water". Users are also allowed to prescribe a fixed water temperature for each simulation domain using the "water_temperature" parameter in the "[settings]" section.

Regarding the limitations of the building height data in the online data sets and in the simulations using online data sets, we agree with the reviewer that the OSM building data sets do not have a reasonable coverage of building heights, at least for the two case studies presented in the manuscript. We have revised this as follows:

Although Berlin_OL captures the building outlines in the simulation domain, the lack of accurate building height leads to an underestimation of $\theta_{2m}$ (Figure 8b) and an overestimation of $WS_{10m}$ (Figure 8j) compared to Berlin_CSD.

Overall, the static driver generated by GEO4PALM and, subsequently, the simulation fidelity of PALM is highly dependent on the input geospatial data quality. With high-quality data, GEO4PALM can create static drivers (Belin_GEO) that are comparable to PALM CSD (Berlin_CSD). Without any local data, GEO4PALM can only represent the simulated environment with limited details as presented in Berlin_OL. The lack of building height information and the coarse resolution of the online data sets may not be suitable for a realistic simulation in the urban environment. These data sets, however, could be useful for coarse simulations, for example, the parent domains of the focused simulation area.

In order the study the effect of the static drivers generated with different tools, the authors present PALM simulations using the respective static driver. Here, however, the description of the simulations is not sufficient enough. Which PALM model version was used (not only list it at the very end of the paper), what atmospheric forcing was applied, what were the most import model run parameters? Furthermore, the authors show only the results of the last time step of a 6 hour simulation. This approach is not useful because of the turbulent nature of the LES model. Even with very tiny modifications, the results of single time steps are expected to differ considerably. Thus, calculate appropriate temporal averages and compare these.

The model and simulation configuration has been added as follows:

We used the PALM model system 22.10 to conduct the simulations. For demonstration purposes, all simulations were initialised with idealised forcing. A northerly was prescribed with a wind speed of 4 m s$^{-1}$. At the initialisation, the surface potential temperature is 295.65 K with no vertical gradient in the first 2000 m from the surface and a vertical gradient of 0.3 K per 100 m for the levels above 2000 m up to the top boundary. The prognostic equation for the water vapour mixing ratio is switched off. Periodic lateral boundary conditions are used with the clear sky radiation scheme. The Radiative Transfer Model (RTM; Krc et al., 2021) and Land Surface Model (LSM; Gehrke et al., 2021) were switched on for all domains. The Urban Surface Model (USM; Resler et al., 2017), and Plant Canopy Model (PCM; Maronga et al., 2020) are only switched on for the child domains (N02 and N03).

For the comparison between simulations, we have calculated hourly averages and compared the results for the 2nd and the last simulation hours. Please refer to the revised manuscript and tracked changes for the added discussions. The figures comparing simulations between GEO4PALM and PALM CSD are presented below:

[Figure]

**Figure 8.** Differences of simulation results (shown in Figures 6 and 7) between GEO4PALM simulations and PALM CSD simulation. The differences are results of GEO4PALM simulations subtracted from the PALM CSD simulation: (a-d) differences in $\theta_{2m}$, (e-h) differences in $T_{sfc}$, (i-l) differences in $WS_{10m}$, and (m-p) differences in $R_{net}$. From left to right, the first and the second columns show the differences between Berlin_OL and Berlin_CSD for the 2nd and 6th hour of the simulations, respectively. The third and fourth columns show the differences between Berlin_GEO and Berlin_CSD for the 2nd and 6th hour of the simulations, respectively. Refer to the panel labels for the corresponding simulation.

[Figure]

**Figure 13.** Similar to Figure 8, but for the Christchurch case. Refer to the panel label for the corresponding simulation.

**Minor issues and comments**

palmpy[1] is another static driver tool that is freely available. In particular, it supports input of geo-referenced data as well. Include this in the paper.

We have included palmpy as follows:

In addition to PALM CSD, palmpy (https://github.com/stefanfluck/palmpy; Fluck, 2020) is another tool developed to generate static driver input for PALM applications at the Center for Aviation (ZAV), Zurich University of Applied Sciences, Switzerland (Liu et al., 2022). The palmpy tool is more generally applicable compared to PALM CSD, but, to the best of our knowledge, it has mainly been applied to regions in Switzerland.

L191: Add a reference for WRF4PALM.

We have revised this as follows:

For example, the static input generated by GEO4PALM was used for Christchurch International Airport to demonstrate applications of the WRF4PALM offline nesting tool (Lin et al. 2021).

L240: palm_csd modifies the terrain height of the child domains to follow the average values of the corresponding subdomain of its parent domain. Afterwards, it adjusted to the PALM levels.

We have revised this as follows:

Berlin_GEO and Berlin_CSD used the same geospatial input, and hence, the topography data for the two cases are identical. It should be noted that PALM CSD offers an option to modify the terrain height of the child domains to follow the average values of their corresponding parent domain, allowing for a smoother transition of the flow at the nested child domains' lateral boundaries. After this modification, PALM CSD can be configured to adjust the topography height onto the simulation domain's vertical levels. Currently, such adjustments are not included in GEO4PALM. GEO4PALM adopts DEM directly and lets PALM itself process and convert topography into the simulation grid. These topography adjustment features are switched off for the comparison between PALM CSD and GEO4PALM presented here.

L245: By default, palm_csd removes vegetation types with high vegetation independent of further input data (and normally replaces it with resolved vegetation). This is done to avoid numerical issues when using a high roughness length with small vertical grid spacing. Please add that.

We have added this as follows:

This is caused by the adjustments applied in PALM CSD. It corrects vegetation type when a vegetation height is available and is indicative of low-laying plant cover. PALM CSD also alters the vegetation type for grid points where LAD data are available, i.e., where the plant canopies are resolved. This is to avoid numerical issues when using a high roughness length with small vertical grid spacing. In addition, a tall vegetation type with high roughness length plus the resolved plant canopies could over-represent the drag of the vegetation. Subsequently, the flow reduction may be overestimated. This feature is currently not available in GEO4PALM.

We have added more discussions on the vegetation type adjustment for the Berlin case:

Adjustments in the vegetation type are evident in the surface variables ($T_{sfc}$ and $R_{net}$; Figures 8g-h and 8o-p), especially at the beginning of the simulations (Figures 8g and 8o). Excluding the water bodies and buildings, the blue patches in Figure 8g coincide with the brown patches in the vegetation types of Berlin_GEO shown in Figure 5b. Without the adjustments in the vegetation type, these vegetation patches in Berlin_GEO lead to a positive bias in $R_{net}$ compared to Berlin_CSD (Figure 8o). In the last simulation hour, the cold biases in $T_{sfc}$ in Berlin_GEO over the vegetation patches are less significant (Figure 8h). Regarding $R_{net}$, the positive biases resemble the LAI patterns of Berlin_GEO shown in Figure 4b, showing the potential impact of inaccurate LAD input. The signal of the adjusted vegetation type, however, is not clear in $WS_{10m}$ as shown in Figures 8k-l. Without the vegetation type adjustment, the wind speed is expected to be lower in Berlin_OL, while we cannot identify such impact from Figures 8k-l. More investigation is required to determine the appropriate adjustment in vegetation type.

And for the Christchurch case:

The differences between Chch_GEO and Chch_CSD agree with the assumption that, without the vegetation type adjustment and with a potentially over-estimated LAD, the simulation will create biases, especially in the simulated winds. These differences are evident in the Christchurch case because of low buildings and a higher proportion of the resolved plant canopies. In the Berlin case, the tallest building is 123.6 m with a domain-averaged building height of 23.9 m. The highest building in the Christchurch case is only 25.0 m high, and the domain-averaged building height is 4.8 m, while most of the trees in the Riccarton Bush are over 12.0 m high. Therefore, in the Christchurch case, the vegetation is the main surface forcing, and such forcing could alter the simulation results significantly.

And discussion of limitations of GEO4PALM:

In addition, GEO4PALM does not apply any adjustments to vegetation types and/or topography. Whether the adjustments would improve the simulations requires further investigation, which is out of the scope of this paper, but should be considered in future developments of GEO4PALM.

L254: The missing building in Berlin is called Berlin Palace [2]. It was recently reconstructed and, thus, not included in the DLR data.

We have revised this as follows:

However, the DLR geospatial data do not include the Berlin Palace (as shown in the centre of Figures 4a, and 5j-5l), while this building is present in Berlin_OL (Figure 5j). This building was recently reconstructed and hence is not included in the DLR data set and subsequently not included in Berlin_GEO and Berlin_CSD.

L271: In which way does palm_csd not support water bodies? Both the definition of the water body type and the water surface temperate is supported.

We have revised this as follows:

This is because $T_{sfc}$ and $R_{net}$ are more sensitive to the differences in the input water temperature, the input building types, adjustments applied in vegetation types, and LAD estimation methods. PALM CSD allows users to provide an input of the spatial distribution of water type and water temperature. Otherwise, it uses the default water temperature of 283.0 K embedded in the code. Similar to PALM CSD, GEO4PALM allows a spatial input of water temperature. However, we do not have water temperature data available. For the Berlin case study, GEO4PALM obtained water temperature from the GHRSST data, which is 275.0 K. For comparison and demonstration purposes, we did not modify the default water temperature in PALM CSD, while users can modify the source code for their simulations. Such a difference in water temperature leads to significant contrasts in $T_{sfc}$ (Figures 8g-h) and $R_{net}$ (Figures 8o-p) between Berlin_GEO and Berlin_CSD.

L349: Remove "6.0". PALM uses years as the version number now as also used in L351.

We have removed "6.0".

**Reply to Anonymous Referee #2**

**General comments**

However, the quality of the available online data used in GEO4PALM does not meet the requirements of a highly detailed simulation, which is usually the aim if someone utilizes PALM. The authors state that the online data are of coarse resolution. When focusing on urban simulations, the focus of many of the recent PALM applications, building-height information are rarely available. Missing this main information about the urban setup, no meaningful results can be expected from the PALM simulation. The free online data, however, could be a handy addition to coarse simulations of the surrounding areas of a focused region. The authors should bring out this aspect more clearly.

We agree with the reviewer that the coarse online data sets in GEO4PALM are not suitable for high-resolution urban applications. We have added discussions for the Berlin case:

Although Berlin_OL captures the building outlines in the simulation domain, the lack of accurate building height leads to an underestimation of $\theta_{2m}$ (Figure 8b) and an overestimation of $WS_{10m}$ (Figure 8j) compared to Berlin_CSD.

Overall, the static driver generated by GEO4PALM and, subsequently, the simulation fidelity of PALM is highly dependent on the input geospatial data quality. With high-quality data, GEO4PALM can create static drivers (Belin_GEO) that are comparable to PALM CSD (Berlin_CSD). Without any local data, GEO4PALM can only represent the simulated environment with limited details, as presented in Berlin_OL. The lack of building height information and the coarse resolution of the online data sets may not be suitable for a realistic simulation in the urban environment. These data sets, however, could be useful for coarse simulations, for example, the parent domains of the focused simulation area.

 Additionally, we have added discussions on such limitations:

The fidelity and features of static driver input depend on the input geospatial data quality and availability, which have been a particular challenge when conducting microscale simulations. GEO4PALM provides several interfaces for users to download global geospatial data sets, which include the basic features of PALM static input, such as topography and land use typology. However, as shown in the case studies, these online data sets are of coarse resolution and are not suitable for high-resolution urban applications at the meter scale. These online data sets could be handy for quickly conducting simulations over a large area with coarse resolutions. The building height and plant canopy information are currently missing in the online data sets covered by GEO4PALM.

In the manuscript, the authors compare different approaches and data sources to create a static driver file for PALM. They compare their tool against PALM's own tool, palm_csd. palm_csd has different checks and modifications of input data included which are based on experience over many simulations (e.g., terrain-height modification for nested setups; adjustment of vegetation types below resolved plant canopies). These include, e.g., the replacement of forest in the land cover data with resolved plant volumes using the leaf-area density. This highly improves simulation results. Such checks are not (yet) implemented in GEO4PALM. This can lead to simulations suffering from already well-known mistakes in the input datasets and might delay research projects. GEO4PALM should therefore be extended by further checks of the input data. The authors are partly aware of this but should state this more clearly in the outlook.

We have clarified this in the outlook as follows:

While PALM CSD applies various checks and modifications of the input geospatial data, e.g., topography and vegetation types, these features are not yet implemented in GEO4PALM. As shown in the Christchurch case study, the adjustments of vegetation type could have a substantial impact on the simulation results. Furthermore, this paper only compared GEO4PALM to PALM CSD, while palmpy and PALM-4U GUI are the other tools that could have more advanced features. Hence, in addition to the investigation and development of vegetation type adjustment, Future work may also include the comparison between the four tools for better improvements on GEO4PALM.

The differences in the static drivers due to the missing adjustments in GEO4PALM compared to palm_csd are mentioned in the manuscript. However, a detailed analysis of the differences in the simulation results due to these differences is missing.

We have added this as follows:

This is caused by the adjustments applied in PALM CSD. It corrects vegetation type when a vegetation height is available and is indicative of low-laying plant cover. PALM CSD also alters the vegetation type for grid points where LAD data are available, i.e., where the plant canopies are resolved. This is to avoid numerical issues when using a high roughness length with small vertical grid spacing. In addition, a high vegetation type with high roughness length plus the resolved plant canopies could over-represent the drag of the vegetation, and subsequently, the flow reduction may be overestimated. This feature is currently not available in GEO4PALM.

We have added more discussions on the vegetation type adjustment for the Berlin case:

[revised manuscript text omitted]

L41: palm_csd can be used for any set of data as long as the user provides the data in netCDF format. This is also stated by Heldens et al. (2020).

The reviewer is correct that PALM CSD can use any data set as long as the data are in NetCDF format. However, this requires significant efforts from the users regarding data pre-processing. We have revised this as follows:

However, the data processing routine provided by Heldens et al. (2020) is heavily dependent on the geospatial data set prepared by the German Space Agency (DLR), for example, for three cities in Germany (Stuttgart, Berlin, and Hamburg) described in their study. The PALM CSD tool can only process data in NetCDF format with its own particular data standard, which requires users to dedicate significant time to pre-processing the geospatial data.

Users are responsible for a significant amount of data pre-processing before using these tools. Therefore, in many other regions, for instance, New Zealand, where only a small number of geospatial data have been prepared by local authorities, big hurdles still exist to apply PALM with realistic land surface characteristics.

L46: The common static-driver preparation tool is palm_csd. This tool is part of PALM and available to every PALM user. The statement made that there is no common tool available is hence incorrect.

The reviewer is correct that PALM CSD is the common static-driver preparation tool. We meant to say the common tool for pre-processing the geospatial data including features like reprojection etc. To clarify this, we have revised this sentence as follows:

Furthermore, the lack of a highly applicable static driver preparation tool likely hinders the reproducibility of scientific results across different regions and research groups.

L46: Section 2 does not describe the PALM input data standard. It just references it and briefly explains what is covered by GEO4PALM.

We have changed the title of Section 2 to "PALM features in GEO4PALM".

L85: Wouldn't "configure file" or "config file" be a better name than "namelist" for the input file? The main executable is even called "run_config_static.py". A "namelist file" refers more to a file containing one or more Fortran namelists.

We agree with the reviewer that "namelist" is a terminology more commonly used in Fortran. Since we are using the Python programming language, we have changed "namelist" to "configuration file" in the manuscript and the GEO4PALM code for clarity.

L92: "The geospacial input data are interpolated or extrapolated..." Are they really extrapolated? Don't you mean averaged? Same in line 93, Figure 2 and other places in the text.

We have revised the wording "interpolated or extrapolated" to "resampled".

L96: Does GEO4PALM automatically convert shapefiles to geotiffs or is this step down to the user? This is not made clear. If shapefiles are not automatically converted, then why not?

Based on our experience, the shapefiles usually have multiple levels and variables. The variable names also vary with the shapefiles. It is exhaustive to include all the features relevant to shapefiles, and therefore, we provide a small conversion script and leave the decisions at the users' end. This has been clarified as follows:

Different from the rasterised GeoTIFF files, the shapefiles are usually vectorised with multiple layers, and each layer has its own designated name, which varies with the data set. Due to the complexity of shapefiles, it is exhaustive to include shapefiles and all the embedded layers as a direct input in GEO4PALM. Therefore, a script shp2tif.py is provided as a GEO4PALM pre-processing tool for users to convert shapefiles to GeoTIFF format of the desired resolution (finest in the input configuration file by default). This script converts one layer of the shapefile at a time allowing users to choose the layer based on their applications.

L116ff: The resolution of the SST dataset (0.01°) is quite coarse compared to the aimed resolution of PALM simulations (mostly in order of a few meters). What happens to grid points where there is no SST available? Are there any assumptions made in GEO4PALM? What happens to small rivers or even ponds and fountains?

We agree with the reviewer that sea surface temperate (SST) data has a coarse resolution and should not be used to map the locations of water bodies. GEO4PALM obtains the water temperature from the nearest grid point of the SST data set. The SST may not be suitable for simulations with fine resolution of, for example, 1 m. It was not explained explicitly in the original manuscript that users can input their own calibrated water temperature files. In addition, we have revised the code and added an option for users to prescribe a fixed water temperature for all water bodies in the simulation domain in case users have a better estimation of the water temperature. These have been clarified and revised as follows:

Users can provide their own water temperature map in a GeoTIFF file or a prescribed water temperature in the configuration file for all water bodies in the simulation domains. Alternatively, users can utilise the online sea surface temperature (SST) data set downloaded by GEO4PALM.

By default, GEO4PALM downloads the version 4.1 Multiscale Ultrahigh Resolution (MUR) of a Group for High-Resolution Sea Surface Temperature (GHRSST) Level 4 analysis provided by NASA Jet Propulsion Laboratory (NASA/JPL, 2015). The water temperature is obtained from the nearest grid point of the SST data set to the PALM simulation domains. To download and use this data set, users must specify "online" in the configuration file for the variable "water". The datetime of the SST data should be specified using the parameter "origin_time" in the "[case]" section. If users have spatial water temperature data available for water bodies in GeoTIFF format, they can specify the data file name in the configuration file for the variable "water". Users are also allowed to prescribe a fixed water temperature for each simulation domain using the "water_temperature" parameter in the "[settings]" section.

L134: A reference to Table B2 is missing.

We have revised the order of the lookup tables. Now, Table B1 is for the NASA data, Table B2 is for the ESA data, and Table B3 is for the New Zealand data. A reference to the corresponding lookup table has been added as follows:

GEO4PALM source code provides a lookup table (Table B1) for the MODIS Land Cover Type Product...

L143: A reference to Table B3 is missing.

This has been revised as follows:

A lookup table (Table B2) for PALM readable conversion is provided in GEO4PALM source code.

L146: The reference comes too late.

This has been revised based on the previous two suggestions and a reference has been added to Table B3 as follows:

In addition to the lookup tables for NASA and ESA data, GEO4PALM source code provides a lookup table (Table B3) for New Zealand land cover database (LCDB) V5.0 (Landcare Research, 2020).

L155: What happens to OSM data that contain land cover classified as vegetation types? Are those ignored? Why are those ignored?

We are aware of the OSM land cover data set. However, we found that the OSM land cover data set does not have good coverage for many regions in the world. For example, for Christchurch and its surrounding areas, approximately 68% of areas have missing/no data (see the screenshot below; white in the pie chart indicates missing/no data).

[Figure]

For the Christchurch central metropolitan area, there still are approximately 15% of missing data (see the screenshot below). This means the OSM land use data set requires further pre-processing and investigation and therefore this is not included in GEO4PALM.

[Figure]

However, for cities with good data coverage, the OSM land cover data set could be a good option. Therefore, we have added discussions as follows:

Note that OSM also provides land use classification, but does not have a good spatial coverage for many regions in the world. Therefore, GEO4PALM currently does not support OSM land cover data. Users are encouraged to use the OSM land cover data set with a modified land use type conversion table like those shown in Appendix B if their PALM simulations are conducted for regions with good spatial coverage of OSM land cover data. GEO4PALM is adaptable to any land use data provided in shapefile or GeoTIFF format.

L162: You already assume that the filter height for vegetation height might not be appropriate for all cases. Then why don't you introduce a configuration parameter that can easily be adjusted in the configuration/namelist file? Such a filter height would depend on the case. But changing the source code will introduce the change to all cases/jobs.

We agree with the reviewer and have added the filter height as a parameter in the configuration file. The corresponding text has been revised as follows:

To avoid noise from other surface geometry, GEO4PALM applies an automatic process in which surface objects with height less than the filter (tree_height_filter in Table 2) are removed such that objects like cars or fences are not included as vegetation. The default value of the filter is 1.5 m, and users can adjust the value in the configuration file (tree_height_filter in Table 2). With high-quality data, this noise filter can be set to a desired low value (≥ 0.0 m) such that low objects like grass, long grass, and bushes, can be included and represented in PALM simulations.

Eq.1: Use upright font for multi-letter variables (LAD). Variable *h* is not explained in the text.

We have corrected the font style for LAD. We have explained the variable $h$ as follows:

where $LAD_m$ is the maximum LAD, $h$ is the tree height, $z_m$ is the height where the LAD reaches $LAD_m$, and $n = 6$ when $z < z_m$ and $n = 0.5$ when $z \geq z_m$.

L167: Should be $LAD_m$ instead of $L_m$. $z_m$ is the height where $LAD = LAD_m$. According to Lalic and Mihailovic (2004), n=0.5 for z ≥ zm.

We have revised this as follows:

where $LAD_m$ is the maximum LAD, $h$ is the tree height, $z_m$ is the height where the LAD reaches $LAD_m$, and $n = 6$ when $z < z_m$ and $n = 0.5$ when $z ≥ z_m$.

L168: The original source of $z_m$ being between 0.2h and 0.4h is Kolic (1978) "Forest Ecoclimatology", University of Belgrade, 295pp. as stated by Lalic and Mihailovic (2004). Please adjust the reference.

We have added the reference as follows:

According to Kolic (1978) and Lalic and Mihailovic (2004), the normalised value of $z_m$ ranges from 0.2 to 0.4 depending on the tree type.

L170: Does the variable 'tree_lai_max' describe the value of $LAD_m$? Then, the naming is incorrect or misleading as it describes the maximum leaf-area density not the maximum index. See also Table 2.

Yes, the variable 'tree_lai_max' describes the value of $LAD_m$. We have revised the naming in Table 2 and the main text as follows:

LAD at each vertical level is derived based on $LAD_m$ and $z_m$. Currently, GEO4PALM only allows users to provide fixed LADm (tree_lad_max in Table 2) and $z_m$ (lad_max_height in Table 2) values as input. GEO4PALM automatically scales the LAD based on the height of the vegetation canopy at individual grid points. This approach does not take account of spatial variation in LAD calculated from different tree species, while it is still useful in cases where no LAD or leaf area index (LAI) data are available. For cases in which LAD or LAI information is available, users are advised to adjust the code to directly read the spatial LAD/LAI information. As Lalic and Mihailovic (2004) stated, LADm can be derived from LAI.

L183: PALM's input and output files contain geospatial information. They are even part of the PALM input/output data standards PIDS and PODS.

We agree with the reviewer that PALM's input and output files contain geospatial information, but the coordinates of latitudes and longitudes may not be accurate. The static driver created by GEO4PALM does not contain all the geospatial information. To clarify this and avoid future confusion, we have revised this as follows:

PALM's own input and output files can contain geospatial information, while the geospatial projection references sometimes may not be included accurately in a NetCDF file. Geospatial coordinates with correct geospatial projection could be important in real-world applications, especially when compare PALM results to observations. To overcome this potential issue, instead of providing geospatial information in the NetCDF files, a GeoTIFF file with coordinate information is created by GEO4PALM along with each static driver. We recommend that GEO4PALM users use the GeoTIFF file to better reference the geospatial coordinate.

L200: palm_csd only includes information about the coordinate reference system for a selection of projections. But the data processing of palm_csd does not depend on any

projection. palm_csd handles any data independent of their projection. Therefore, it is totally capable of handling the input data of the Christchurch case.

We agree with the reviewer and have processed Christchurch data for PALM CSD. A new simulation was conducted for the Christchurch case using PALM CSD. We have clarified this as follows:

At present, PALM CSD only supports ten UTM projections between zone 28N and zone 37N or a pre-processed rectangle map projection. Therefore, for Otautahi / Christchurch, we pre-processed all the local data to the map projection of EPSG:2193 (New Zealand Transvers Mercator) using geographic information system (GIS) software and tools in GEO4PALM. Then, the corresponding GeoTIFF files were processed into PALM CSD compatible NetCDF files.

L232: palm_csd can consider a high amount of information for vegetation processing. However, not everything is required. Assumptions can also be made if input data is not available. GEO4PALM, however, only supports the simplified method as stated in the text. Modifying the source code to meet the user's needs is of course also true for palm_csd. Please modify this part to better compare both tools.

We have revised this as follows:

A lot of data processing is required to have vegetation data pre-processed for PALM CSD. For the estimation of LAD, at least the vegetation height, vegetation type, and LAI data are required. Considering the inconsistency in data sources and quality worldwide, GEO4PALM adopts the simplified method to calculate LAD. Both GEO4PALM and PALM CSD can be modified further by users depending on their modelling needs.

By the time the manuscript was submitted, we did not have any LAD data available locally. We have had a field campaign to obtain LAD data for PALM simulations, and we aim to add a better LAD estimation based on PALM CSD in the future. We have clarified this in the outlook as follows:

GEO4PALM currently only accepts vegetation heights as input for plant canopy due to a lack of geospatial data. In the future, we aim to improve this feature based on the PALM CSD tool (Heldens et al., 2020) and to include more data sources if available.

L238: palm_csd does modifications of the terrain height based on recommendations for the internal nesting module. It removes differences of the terrain height between child and parent domain and allows for a smoother transition of the flow at the child's domain border. This modification, however, is optional and can be turned off in the configuration file. The modification of the terrain height in palm_csd is recommended because PALM itself does not do this. So, the statement in line 240 is incorrect. For a better comparison of the static driver from GEO4PALM and palm_csd, the terrain-height correction of palm_csd should have been turned off.

Based on the reviewer's suggestion, we have turned off the topography adjustment features and conducted Berlin_CSD again for a better comparison between PALM CSD and GEO4PALM. Section 4.2 has been revised based on the new results. We have also revised the topography adjustment discussion as follows:

Berlin_GEO and Berlin_CSD used the same geospatial input, and hence, the topography data for the two cases are identical. It should be noted that PALM CSD offers an option to modify the terrain height of the child domains to follow the average values of their corresponding parent domain, allowing for a smoother transition of the flow at the nested child domains' lateral boundaries. After this modification, PALM CSD can be configured to adjust the topography height onto the simulation domain's vertical levels. Currently, such adjustments are not included in GEO4PALM. GEO4PALM adopts DEM directly and lets PALM itself process and convert topography into the simulation grid. These topography adjustment features are switched off for the comparison between PALM CSD and GEO4PALM presented here.

L248: If a forest area is represented in the static driver by the vegetation type and also by the leaf-area density, it will be over-represented in the PALM simulation. The drag of the forest is parameterized in the roughness length used for the vegetation type but also the resolved plant canopy introduces drag. Hence, the flow reduction is overestimated. This is why palm_csd alters the vegetation type below resolved plant canopies. These differences are not further investigated in detail when comparing the simulation results.

We have added more discussions in the revised manuscript, and we have revised the vegetation adjustment discussions as follows:

This is caused by the adjustments applied in PALM CSD. It corrects vegetation type when a vegetation height is available and is indicative of low-laying plant cover. PALM CSD also alters the vegetation type for grid points where LAD data are available, i.e., where the plant canopies are resolved. This is to avoid numerical issues when using a high roughness length with small vertical grid spacing. In addition, a tall vegetation type with high roughness length plus the resolved plant canopies could over-represent the drag of the vegetation. Subsequently, the flow reduction may be overestimated. This feature is currently not available in GEO4PALM.

L254: should be "present" instead of "presented".

This has been corrected.

L254: The Humbold Forum was not yet constructed when the data were gathered by the DLR.

We have revised this as follows:

However, the DLR geospatial data do not include the Berlin Palace (as shown in the centre of Figures 4a, and 5j-5l), while this building is present in Berlin_OL (Figure 5j). This building was recently reconstructed and hence is not included in the DLR data set and subsequently not included in Berlin_GEO and Berlin_CSD.

L256: I recommend to write $\theta_{2m}$ ("2m" in subscript) similar to $T_{sfc}$. The same goes for $WS_{10m}$ (wind speed at 10 m height).

We have revised these throughout the manuscript as suggested by the reviewer.

L271: palm_csd has an option to set the water surface temperature based on the water type. So, the statement made is incorrect. Besides this, the user is free to adjust palm_csd to also read water surface temperatures from an input raster file and output this to the static driver.

We have revised this as follows:

PALM CSD allows users to provide an input of the spatial distribution of water type and water temperature. Otherwise, it uses the default water temperature of 283.0 K embedded in the code. Similar to PALM CSD, GEO4PALM allows a spatial input of water temperature. However, we do not have water temperature data available. For the Berlin case study, GEO4PALM obtained water temperature from the GHRSST data, which is 275.0 K. For comparison and demonstration purposes, we did not modify the default water temperature in PALM CSD, while users can modify the source code for their simulations.

L281: palm_csd is not dependent on map projection. The tool does not use any map projection but treats input raster files as simple raster arrays. Therefore, the actual projection is irrelevant for palm_csd. The only thing that is assumed is a rectangle map projection where coordinates are measured in meter. Hence, the input files for Christchurch could have been handled by palm_csd. For a better comparison of GEO4PALM and the existing palm_csd tool, the Christchurch case should also be covered with palm_csd.

Based on the reviewer's suggestion, we have processed all the input data for the Christchurch case. We have used PALM CSD to create static drivers for the Christchurch case and named the simulation Chch_CSD. We have revised the Christchurch case study sections and added analysis and discussions with the addition of Chch_CSD.

L311f: What is the meaning of the last two sentences of this paragraph? What mismatch? Of course, if you use different data sources, data might not agree to each other, especially if the data sources refer to different points in time (e.g., newly constructed buildings in OSM, land-cover data still shows forest area). And in what way do the results agree with those shown by Heldens et al. (2020)?

We agree with the reviewer that these sentences do not deliver clear information. As stated by the reviewer, different data sources may be obtained at different times. For clarification, we have revised this for the Christchurch case specifically:

The OSM data used in Chch_OL may be more up-to-date, while Chch_GEO used local data that were processed and checked manually. In GEO4PALM, the OSM interface recognised the footpath on the Riccarton Mall rooftop parking lot as pavements. GEO4PALM removes the buildings if they overlap with pavements, whereas PALM CSD removes the pavements when they overlap with buildings. More investigation is needed to determine a more appropriate overlap check.

Our simulation results show that the surface structure has a strong impact on surface temperature and net radiation, similar to the results shown by Heldens et al. (2020). This statement is trivial, and hence, we decided to remove these sentences.

L321: As noted above, palm_csd is widely applicable and also shipped with PALM. So, it is available to the entire PALM community. Besides this, there is another tool called palmpy (https://github.com/stefanfluck/palmpy) that is also capable to handle even vector data like ESRI shapefiles.

We have revised the Conclusions section as follows:

To the best of our knowledge, current static driver generation tools in the PALM community (PALM CSD, palmpy, and PALM-4U GUI) either heavily rely on users to pre-process geospatial

data, or have not been applied to many regions in the world, for that case, for example, New Zealand.

This paper only compared GEO4PALM to PALM CSD, while palmpy and PALM-4U GUI are the other tools that could have more advanced features. Future work may also include the comparison between the four tools for better improvements on GEO4PALM.

L330: Building-height information are often available from the local authorities. For research purposes, these datasets might also be free or at a discount compared to commercial uses.

We agree with the reviewers that some local authorities could provide building height information. This may be prevalent for large cities, for instance, in Europe, but this is not true for, for example, cities in New Zealand and/or Australia. New Zealand local authorities do not usually have the building height information, and if, in a very rare case, they have such data, it is usually expensive, even for research purposes. We have revised this as follows:

Many regions worldwide do not have high-resolution geospatial data to realise simulations with high fidelity. High-resolution building height data are important for urban applications, but they are not widely available, making simulations over human settlement difficult. While building height information could sometimes be obtained from the local authorities in large cities, the availability of such data sets might vary depending on the regions of interest. Additionally, for research purposes, there might be differences in accessibility and cost, particularly in smaller or less urbanised areas.

L338ff: Apart from the coarse classification of settlement areas, the resolution also plays a crucial part. Even if you manage to distinguish settlement areas between highly dense built-up areas, medium dense and sparsely built areas, this is not enough for a PALM simulation with below 10 m resolution. In this case, you need to distinguish between sidewalks, streets, paved backyards, flowerbeds, private pools, etc. This is especially true if you want to analyze your simulation data at the street level or even want to compare your data with measurements. So, the data availability and data quality are a huge bottleneck for high-resolution urban simulations.

We agree with the reviewer that data availability and data quality are the biggest hurdles for high-resolution urban simulations. We have acknowledged the limitation of resolution in the online data sets provided by GEO4PALM:

GEO4PALM provides several interfaces for users to download global geospatial data sets, which include the basic features of PALM static input, such as topography and land use typology. However, as shown in the case studies, these online data sets are of coarse resolution and are not suitable for high-resolution urban applications at the meter scale.

We have cited another building height data set that could be potentially useful for urban simulations:

Using the deep neural network, the GLObal Building heights for Urban Studies (GLOBUS) gives a novel Level of Detail-1 (LoD-1) building data set (Kamath et al., 2022).

L349: Remove "6.0". You are using PALM 22.10.

We have removed "6.0".

L356: Why did you not include the input files for the Berlin case?

We only provided files for one case for demonstration purposes. We have included the input files for the Berlin case in the supplements.

L414: Should be "input for street type".

This has been corrected.

L419: "lad_max_height" describes the height where LAD = $LAD_m$

We have revised this as follows:

tree_lad_max    - input value for maximum leaf area density (LADm)
lad_max_height   - input value for the height where the leaf area density (LAD) reaches LADm

Table 1: Caption: The second sentence should read: "For more detailed descriptions of the variables refer to...".

This has been corrected.

Table 2: "dem" is described as topographical height. However, I guess you mean a digital terrain model, or DTM. A DEM, or digital elevation model, includes the terrain height but also buildings and vegetation. However, PALM specifically requires the terrain height to then place buildings and vegetation on top of that. Please also double-check the text.

We want to clarify the terminology here that a digital surface model, or DSM, includes the terrain height but also buildings and vegetation. A DEM or DTM should be the one that includes the terrain height but excludes buildings and vegetation.

We have clarified the differences between a DEM and a DSM as follows:

One possible solution to obtain vegetation height is to calculate surface objects' height using digital surface model (DSM) and DEM. In addition to the information on gro ground surface altitude contained in DEM, DSM supplies the heights of all surface objects, such as buildings and trees.

We have pointed out the potential issue with the NASA DEM data as follows:

It should be noted that the NASA topography data may have included surface objects' heights (compare Figure 9d to Figures 9e-f), while the DEM and/or topography in the concept of GEO4PALM and PALM refer to the topographical height only. Users should take extra care if they use the NASA topography data.

Figure                                                                                          2:

- Yes/no lines of first decision box do not start at the left/right corners.

This has been corrected.

- From the flowchart it seems that GEO4PALM either downloads all input files or only reads local tif files.

GEO4PALM accepts any combination of downloaded online data and local tif files. This has been revised.

- Why does the box "start to process downloaded and/or user provided geospatial files" have round edges?

The box has been corrected to a rectangular box without round edges. See below for the revised figure.

[Figure]

Figure 6: Are the shown output data instantaneous data or are they averaged over time? Non-averaged data are most likely to be different between the simulations. Due to the many small differences in the static drivers, the turbulence might most certainly develop differently in the simulations. So, differences in the instantaneous data are most certainly to occur. Time-averaged output, especially for the wind speed, should be a better measure to identify differences in simulation results than instantaneous data.

We have calculated hourly averages for all the simulations and have compared GEO4PALM simulations to PALM CSD simulations for the 2nd and last simulation hours. The case studies sections have been revised, and more figures have been added.

Figure 7: The terrain height for the Chch_OL case seems to include vegetation height. So, it might actually be a digital elevation model including the height of the trees. This is obvious when you compare the height information at the park location.

We agree with the reviewer that the DEM obtained from the NASA online database may include the height of the trees. We have added discussion on this issue:

It should be noted that the NASA topography data may have included surface objects' heights (compare Figure 9d to Figures 9e-f), while the DEM and/or topography in the concept of GEO4PALM and PALM refer to the topographical height only. Users should take extra care if they decide to use the NASA topography data.

Figure 9: See comment on Figure 6.

These figures have been revised.

Comment on the usage of the program (appendix A):
Why do you have to alter the source code (utils directory) when using a different lookup table for the landuse? It would be better if you could add a lookup table to the input directory and the tool uses this then. This way you do not have to change (or check) the utils directory every time you switch to another case study.

We agree with the reviewer and have revised the source code such that now users only need to provide the lookup table and specify the filename in the configuration file. Table 2 and Appendix A has been revised to reflect this change.

**Technical corrections**
- The text, including the Figure and Table captions, often misses the articles "the" and "a", e.g. in Table 1, description of "building_type": Required for buildings in *the* urban surface model.

We have gone through the manuscript and revised the articles where applicable.

- Use hyphens for compound words like "land-surface characteristics" (l. 43).

This has been corrected. We have gone through the manuscript and revised the compound words where applicable.

- L63: "Here we describe and document and GEO4PALM toolkit". The second "and" should be "the".

This has been corrected.

- L104: The text goes past the end of the line.

This is formatted automatically by LaTeX, and we have applied several fixes to avoid this.

- L117: "...which can have *an* impact on..."

This has been corrected.

- L165: "...is based on *an* equation..."

This has been corrected.

- Table 2: The writing "metres" is used, however, throughout most of the manuscript "meters" is used.

For consistency, we have replaced "meters" with "metres" in the manuscript.

- L214: "...with *the* finest grid spacing at 3 m, while GEO4PALM can generate *a* static driver with grid *spacings* finer than..."

This has been corrected.

- L257: Use an upright font for multi-letter variables like "WS" (as you already did with LAD). Also, use an upright font for the subscript of $T_{sfc}$ $R_{net}$ etc.

We have corrected the formatting for all the variables.

- L260: Space missing: "2m"

This has been corrected.

- L324: text passes the end of line.

This has been corrected.

- L428: Should be "python" (lower case).

This has been corrected.

---

## Author Response (AR2)

**Note to editor:**

We found that there was a miscommunication between the authors that the leaf area density input variables for GEO4PALM should be the leaf area index (tree_lai_max) rather than the leaf area density (tree_lad_max).

We have used the integral form of the equation listed in the manuscript (Eq.1) and indicated this input variable should be the leaf area index in the original version. During the first round of revision, one of the reviewers pointed out that this should be the leaf area **density**. We followed this suggestion and later found that this variable should be the leaf area **index**. This mistake was only found after the first round of revision was done and hence, we have this issue communicated here.

The only changes here are the names of the variables used in GEO4PALM. GEO4PALM uses the integral form of the leaf area density, which should be the leaf area index. The GEO4PALM code has not been affected and remains the same.

We have corrected the name of this variable in the manuscript and revised the corresponding texts to clarify this. The rest of the manuscript has remained the same.